# Motif-based models accurately predict cell type-specific distal regulatory elements

Paola Cornejo-Páramo[1,2], Xuan Zhang [1], Lithin Louis[1], Zelun Li[1], Yihua Yang[1] & Emily S. Wong [1,2] ✉

Deciphering how DNA sequence specifies cell-type-specific regulatory activity is a central challenge in gene regulation. We present Bag-of-Motifs (BOM), a computational framework that represents distal cis-regulatory elements as unordered counts of transcription factor (TF) motifs. This minimalist representation, combined with gradient-boosted trees, enables the accurate prediction of cell-type-specific enhancers across mouse, human, zebrafish, and *Arabidopsis* datasets. Despite its simplicity, BOM outperforms more complex deep-learning models while using fewer parameters. We validate BOM's predictions experimentally by constructing synthetic enhancers from the most predictive motifs, demonstrating that these motif sets drive cell-type-specific expression. By providing direct interpretability and broad applicability, BOM reveals a highly predictive sequence code at distal regulatory regions and offers a scalable framework for dissecting cis-regulatory grammar across diverse species and conditions.

A fundamental goal of biology is to understand how genetic information is translated into the diverse phenotypes observed across organisms. At the heart of this process lies an intricate regulatory network that controls gene expression with remarkable spatial and temporal precision. Cis-regulatory elements (CREs)—including promoters, enhancers, insulators, and silencers—coordinate the recruitment of transcription factors (TFs) and cofactors to DNA. Enhancers are of particular importance as they act as integrators of developmental and environmental cues to establish and maintain cell identity[1–3]. These elements are can be located far from their target genes and contain clusters of short TF binding motifs (~8–10 bp)[4], whose number can influence the strength of TF binding[5].

Advancements in genome-scale experimental profiling of the regulatory genome have enabled researchers to map CREs across diverse mammalian cell types systematically. This progress has been made possible by genome-wide techniques such as ATAC-seq (assay for transposase-accessible chromatin with sequencing) and ChIP-Seq (chromatin immunoprecipitation with sequencing) of TF binding sites and histone marks that are indicative of enhancer activity. In recent years, single-nucleus ATAC (snATAC-seq) sequencing had provided a data-rich opportunity to study gene regulatory mechanisms across a variety of cell types.

However, understanding the complex regulatory code governing cell-type-specific gene expression remains a central challenge, as the relationship between distal regulatory sequence and cell context is often difficult to capture by sequence alone. The sheer number of potential regulatory elements in large genomes, coupled with the degeneracy and ubiquity of TF binding motifs, makes sequence-based prediction difficult. Many studies use TF motif enrichment[6–10], using tools like FIMO and HOMER, to scan for over-represented motifs using position weight matrices. Motif-based tools for comparing two or more sets of regulatory elements encompass a variety of computational approaches, each differing in purpose, algorithm, and interpretability (e.g. Supplementary Table 1). Statistical enrichment tools such as ChromVAR[8] and MEDEA[9] quantify motif variability or compare accessible regions to reference panels, but they do not model motif combinations or provide model performance metrics on how well a model generalizes. Linear models like IMAGE[7] and ISMARA[6] are computationally efficient but assumes additivity of motif contribution, while ensemble models such as Gimme Maelstrom improve differential motif detection but lack per-sequence interpretability.

[1]Victor Chang Cardiac Research Institute, Darlinghurst, NSW, Australia. [2]School of Biotechnology and Biomolecular Sciences, UNSW Sydney, Sydney, NSW, Australia. ✉e-mail: e.wong@victorchang.edu.au

K-mer–based classifiers (gkmSVM[11], LS-GKM[12]) can discover novel sequence patterns but require additional motif annotation[13,14]. Deep-learning models, including BPNet[15], ChromBPNet[16], and Enformer[17], can learn longer-range dependencies and can achieve high predictive accuracy, but they are computationally intensive, often require large training datasets, and demand specialized tools for interpretation; moreover, some struggle to model the influence of distal enhancers on gene expression[18].

Here we describe BOM, an enumerative motif-based method that addresses these challenges by representing each distal regulatory sequence as an unordered set of motif counts, independent of motif order, orientation, or spacing. Using gradient-boosted trees, BOM captures the combinatorial contributions of TF motifs while remaining computationally efficient. Importantly, the motif-centric representation yields a vocabulary of regulatory features that is biologically interpretable and can be linked directly to gene expression and TF expression. Across diverse datasets spanning multiple species and tissues, BOM achieves high predictive accuracy and outperforms deep-learning models. We further show that synthetic enhancers assembled from the most predictive motifs drive cell-type-specific expression, validating BOM's predictions experimentally. Together, these findings reveal a highly predictive sequence code at vertebrate distal regulatory regions and demonstrate the utility of BOM for dissecting the cis-regulatory logic underlying cell identity.

## Results

### BOM predict cell-state-specific distal cis-regulatory elements

Building on motif-centric methods, we developed a flexible bag-of-motif strategy in which each cis-regulatory element is represented by a vector of motif counts. This minimal representation can be applied to any pair of chromatin-defined conditions (e.g. cell states defined by differential accessibility or histone marks)[19–23]. Classification and regression tasks were performed using the XGBoost gradient-boosting algorithm[24], and SHAP values were used to quantify the contribution of each motif to individual predictions (Fig. 1a).

We benchmarked BOM on single-nucleus ATAC-seq data from mouse embryos at stage E8.25 covering 17 annotated cell types (Fig. 1b)[25]. Candidate CREs were defined as distal (>1 kb from the transcription start site (TSS)), non-exonic peaks and trimmed to 500 bp windows, yielding 12,079 non-overlapping sequences (Supplementary Table 2). For each sequence, motifs were annotated using GimmeMotifs[10,26], a database of clustered TF binding motifs that reduces redundancy (Fig. 1a, Supplementary Fig. 1). Each sequence was encoded as an unordered vector of motif counts ('bag'), forming the input matrix for training and evaluation. On average, ~89% of CREs were annotated by motifs.

Models were trained on 60% of the data, with 20% reserved for validation and 20% for testing. Background sequences were balanced across cell types for binary classifications. Across 2489 held-out sequences, BOM correctly assigned 93% of CREs to their cell type of origin, with average precision, recall and F1 scores of 0.93, 0.92 and 0.92 (auROC = 0.98; auPR = 0.98) (Fig. 1c, d; Supplementary Table 3). Performance was stable across multiple random train–test splits (Supplementary Fig. 2a, Supplementary Table 4). A multiclass model trained jointly on all 17 cell types achieved a precision of 0.99, recall of 0.88 and F1 of 0.93 (auPR = 0.99) (Supplementary Fig. 3; Supplementary Tables 5, 6). To understand model specificity, we tested the same classifier after adding in negative sequences that flank CREs (±2 kb). The model showed a false positive rate of 0.01–0.29 (Fig. 1e; Supplementary Fig. 2c; Supplementary Table 7). To examine sensitivity to training set size, we subsampled endothelial CREs from 30 to 1730 sequences and trained binary classifiers against balanced backgrounds. Matthews correlation coefficients (MCC) remained above 0.7 for sample sizes >330; and even with 30 positives the model achieved an MCC of 0.70 (Supplementary Fig. 2b; Supplementary Table 8).

For following analyses we have restricted to conditions with at least 100 CREs.

During embryonic development, cells can be viewed as a continuum of transient intermediate states. As such, we applied BOM to decipher intermediate cell states across 93 latent cell states ($n$ = 76,607 CREs)[27]. Despite finer state granularity, the models achieved a mean auPR of 0.86, F1 of 0.71 and MCC of 0.74 (Supplementary Fig. 2d, e; Supplementary Table 9), demonstrating that motif composition could mostly capture transitions along developmental trajectories. We further trained multilabel models to annotate CREs active in multiple cell types. Among 15,395 pleiotropic elements, performance dropped markedly (mean auPR = 0.44; F1 = 0.30; MCC = 0.32; Fig. 1f). This suggests that broadly active elements were enriched for ubiquitous housekeeping motifs and relied more on chromatin context or higher-order interactions than on distinctive motif combinations.

To test for generalization across independent datasets, models trained on E8.25 data were used to classify CREs from a snATAC-seq dataset at E8.5[28]. Restricting to 15 matched cell types, the transferred models achieved a mean auPR = 0.85, auROC = 0.85 and MCC = 0.53 (Fig. 1g; Supplementary Table 10), indicating that a motif code learned at one time point can predict cell-type identity across closely related developmental stages.

### BOM outperforms other methods in predicting cell-type-specific CREs

To benchmark BOM against existing sequence-based classifiers, we compared its performance with LS-GKM[12], DNABERT[29], and Enformer[17] on the same distal regulatory element classification task. LS-GKM is a gapped k-mer support vector machine, DNABERT is a transformer-based language model trained on k-mer representations of DNA, and Enformer is a hybrid convolutional–transformer architecture that uses self-attention to model long-range interactions up to 196 kb. All models were evaluated in pairwise (binary) classification across the 17 E8.25 cell types using balanced positive and negative sets. We fine-tuned DNABERT and Enformer on the training data and performed hyperparameter optimization (Methods). Across cell types, BOM achieved a mean area under the precision–recall curve (auPR) of 0.99 and an MCC of 0.93, outperforming LS-GKM, DNABERT, and Enformer by 17.2%, 55.1%, and 10.3% in auPR, and by 77.5%, 211.9%, and 33.4% in MCC, respectively (Fig. 2a, b, Supplementary Table 3).

We next assessed multiclass performance using three simple convolutional neural architectures trained on DNA sequences: (i) a four-layer convolutional neural network (CNN) with fully connected layers (similar to DeepSTARR[30]), (ii) a CNN followed by a bidirectional LSTM (analogous to DeepMEL/DanQ[31,32]), and (iii) Basset which comprises of a three-layer CNN with two fully connected layers[33]. The first two architectures were sequence-based CNNs that were constructed and trained solely on our training dataset, while we fine-turned the Basset model. For each architecture, the output layer predicted all 17 cell types, the training data was augmented with reverse complements of each CRE, and we conducted a hyperparameter search to identify the most performant model. BOM surpassed all deep learning baselines in multiclass classification (Supplementary Fig. 4b, c). In particular, recall for the CNN-based ranged from 0.0 to 0.5, whereas BOM achieved an average precision of 0.98 and recall of 0.88 (Supplementary Table 6).

Finally, we examined whether BOM's fixed 500 bp input window may limit performance on very large cis-regulatory elements such as super-enhancers. We stratified CREs by their overlap with super-enhancers, identified using H3K27ac density profiles from heart and forebrain tissues. Classification accuracy was similar for super-enhancer and non-super-enhancer elements (Supplementary Fig. 5, Supplementary Table 11), indicating that BOM's predictive power is robust to CRE length.

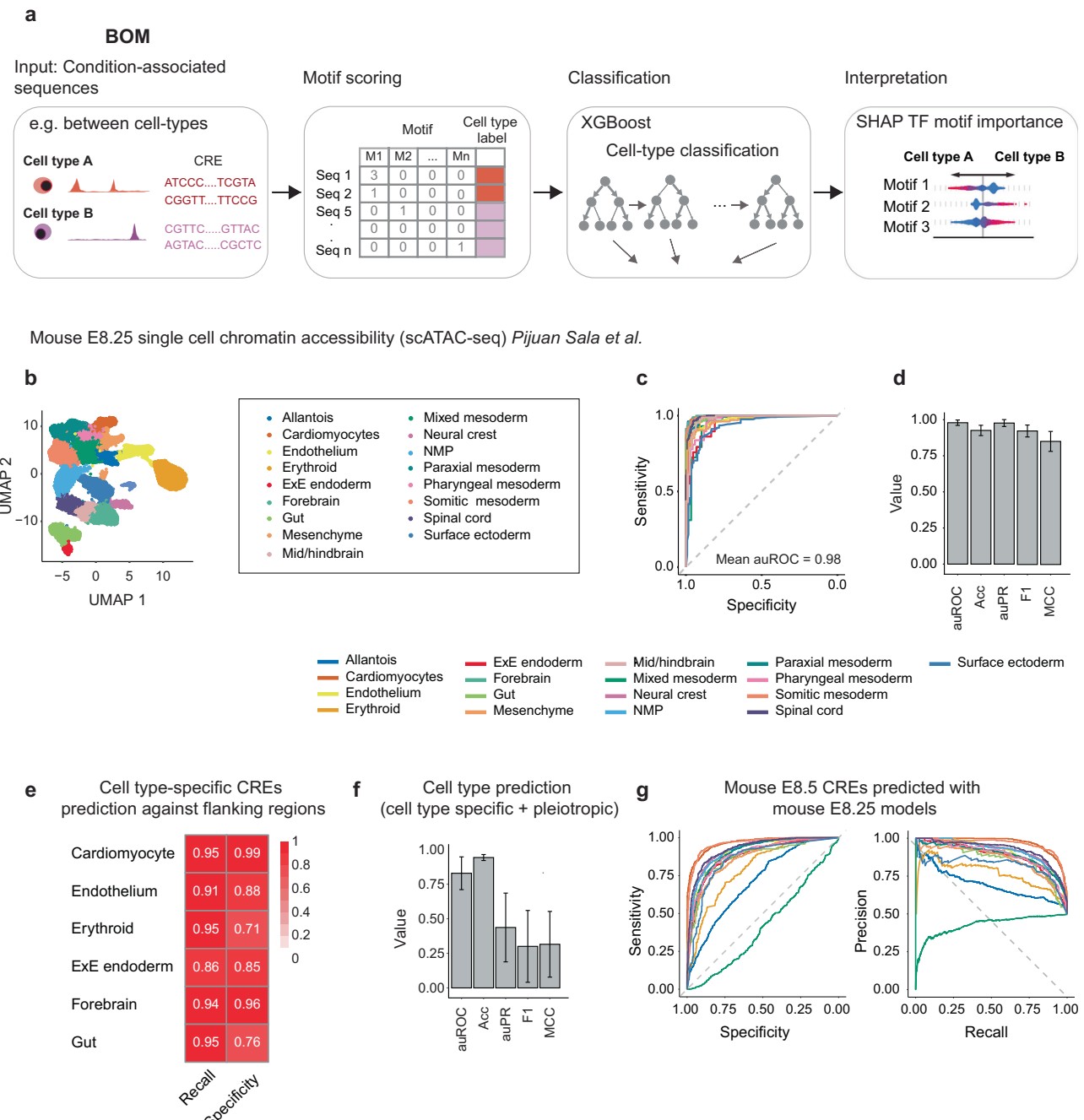

**Fig. 1 | BOM accurately classifies mouse embryonic context-specific CREs.**
**a** Framework of BOM. First, we defined at least two sets of CRE sequences specific to different cell types or conditions. Then, we identify TF binding motif instances within these sequences. Vertebrate motifs from GimmeMotifs were used to annotate CREs. The model is trained to classify cell states via XGBoost for binary or multiclass classification. SHAP values are calculated to explain the importance of TF binding motifs for the classification task. **b** Overview of the mouse E8.25 snATAC-seq dataset from ref. 25. UMAP shows the cell types identified in the mouse embryos and used in our analyses. Notochord cells were excluded because of the low number of CREs. **c** Receiver operating characteristic (ROC) for the binary classification of each of the cell types in the mouse E8.25 embryonic dataset. The mean value of AUC is shown. **d** Summary of the prediction statistics of binary BOM models trained to classify cell-type-specific CREs for 17 cell types (auROC area under the receiver operating characteristic (ROC) curve, Acc accuracy, auPR area under the precision recall (PR) curve, F1 F1 score, MCC Matthews correlation coefficient). The error bars represent standard deviation. **e** Recall and specificity for the classification of cell type-specific CREs and CRE-flanking regions as the negative class **f** Summary of the prediction statistics of BOM models trained to classify cell type-specific and pleiotropic CREs across 18 cell types. Error bars are standard deviation. **g** ROC curves (left) and precision-recall curves (right) for the classification of mouse E8.5 CREs using the models trained on mouse E8.25 CREs. Source Data are provided as a Source Data file.

## Lenient motif detection thresholds improve performance
Selecting the appropriate threshold for motif scanning with position weight matrices is inherently challenging[34,35]. To assess its impact, we scanned motifs in distal elements using FIMO[26] at three $q$-value cutoffs: 0.1, 0.3 and 0.5. Stricter thresholds reduced predictive power on models trained from human cell lines (Supplementary Fig. 6, Supplementary Table S12), whereas at $q \leq 0.5$ about 83 % of the nucleotides in each CRE were annotated as motifs. Thus, a substantial fraction of bases within CREs encode features useful for classification.

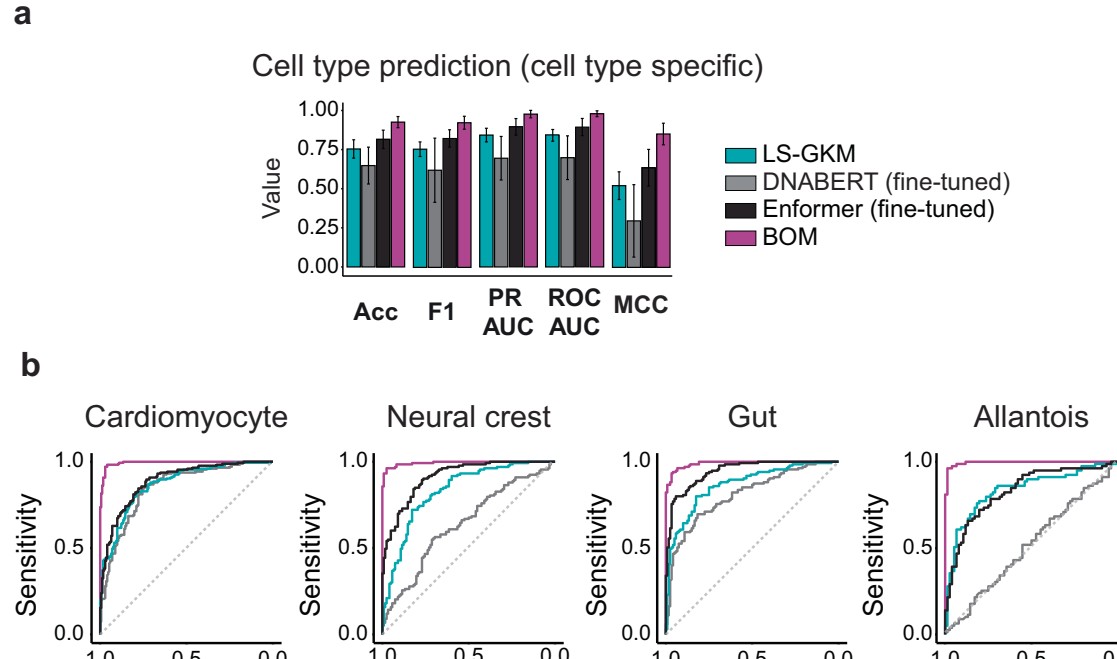

Mouse E8.25 single cell chromatin accessibility (scATAC-seq) *Pijuan Sala et al.*

**Fig. 2 | BOM outperforms other methods for CRE binary classification.**
**a** Comparison of cell-type-specific scATAC-seq peaks binary classification across 17 mouse embryonic cell-types using BOM, DNABERT (fine-tuned), Enformer (fine-tuned), LS-GKM. Acc accuracy, F1 F1 score, auPR area under the PR curve, auROC area under the ROC curve, MCC Matthews correlation coefficient). Error bars represent the standard deviation (centered on the value). **b** ROC curves for four cell types. Models are color-coded as in (**a**). Source Data are provided as a Source Data file.

This observation was supported by analysis of bioChIP-seq data for six cardiac transcription factors (Mef2a, Mef2c, Nkx2-5, Srf, Tbx5 and Tead1)[36]. Mean ChIP signal at motif summits indicated that $q$-value cutoffs above 0.1 were necessary to detect binding sites for Nkx2-5 and Tead1 (Supplementary Fig. 7). Concordantly, reducing motif counts by only 10 % through subsampling decreased model sensitivity by more than 50% (Supplementary Fig. 8), implying that lenient thresholds retain degenerate or low-affinity motifs, and potentially secondary or non-canonical motifs, that contribute disproportionately to predictive performance.

Finally, we evaluated the effect of eliminating overlapping motif annotations. We sorted motifs by genomic position and iteratively removed overlaps by discarding the lower-scoring motif or choosing randomly when scores were equal. Removing overlapping motifs reduced auROC by 0.32 (Supplementary Fig. 9, Supplementary Table 13), indicating that seemingly redundant annotation may reflect biologically meaningful information. Overlapping motifs could have captured composite binding sites formed by cooperative TF pairs, which differed from the recognition sequences of individual factors, which enhanced model performance[37].

**Distal regulatory motifs distinguish cell context across species**
Beyond mouse data, we applied BOM approach to classify cis-regulatory elements across diverse biological contexts and species. All models were trained using classed-balanced data with a stratified negative set so that cell states with low CRE numbers were not underrepresented (Supplementary Table 2).

We classified 167,148 candidate enhancers across six human ENCODE[38] cell lines defined as enhancers based on histone marks by ChromHMM[38]. BOM achieved mean F1 = 0.92, auPR = 0.98 and

auROC = 0.98 (Fig. 3a, Supplementary Table 14). In snATAC-seq data from human blood and bone marrow spanning 22 cell types ($n$ = 12,796), BOM obtained F1 = 0.90, auPR = 0.95 and auROC = 0.96. Cross-application of each cell-type-specific model to the other cell types showed negligible cross-predictions (Fig. 3b, Supplementary Table 15), indicating that BOM learned distinct cis-regulatory codes for each lineage.

Applying BOM to zebrafish adult tissues, we classified 135,763 distal, non-exonic accessible elements across 11 tissues with F1 = 0.96, auPR = 0.99 and auROC = 0.99 (Fig. 3c, Supplementary Table 16). In Drosophila S2 cells, we trained BOM on a subset of STARR-seq enhancers that had either housekeeping or developmental preference[39]. Using only 15.6% of the data used to train the DeepSTARR[30] model, BOM achieved Pearson correlations of 0.78 and 0.42 under housekeeping and developmental promoters, respectively, compared with DeepSTARR's 0.74 and 0.68 (Fig. 3d). Thus, while BOM outperformed DeepSTARR in the housekeeping context, it was less effective for developmental enhancers, possibly reflecting the small training set and increased complexity of tissue-specific regulatory grammars. In *Arabidopsis thaliana*, we trained a multiclass model to classify open chromatin regions across four root cell types (cortex/endodermis precursors, two endodermal clusters and xylem)[40]. Despite fewer than 300 CREs per class, BOM accurately recovered cell-type-specific regulatory elements (Fig. 3e, Supplementary Table 17, Supplementary Fig. 10).

We next investigated whether BOM could discriminate between tumor and healthy states. Using ATAC-seq peaks that were differentially accessible between blast cells from acute myeloid leukemia (AML) patients and their matched preleukaemic haematopoietic stem cells (HSCs)[41], BOM classified test CREs with F1 = 0.99. SHAP analysis

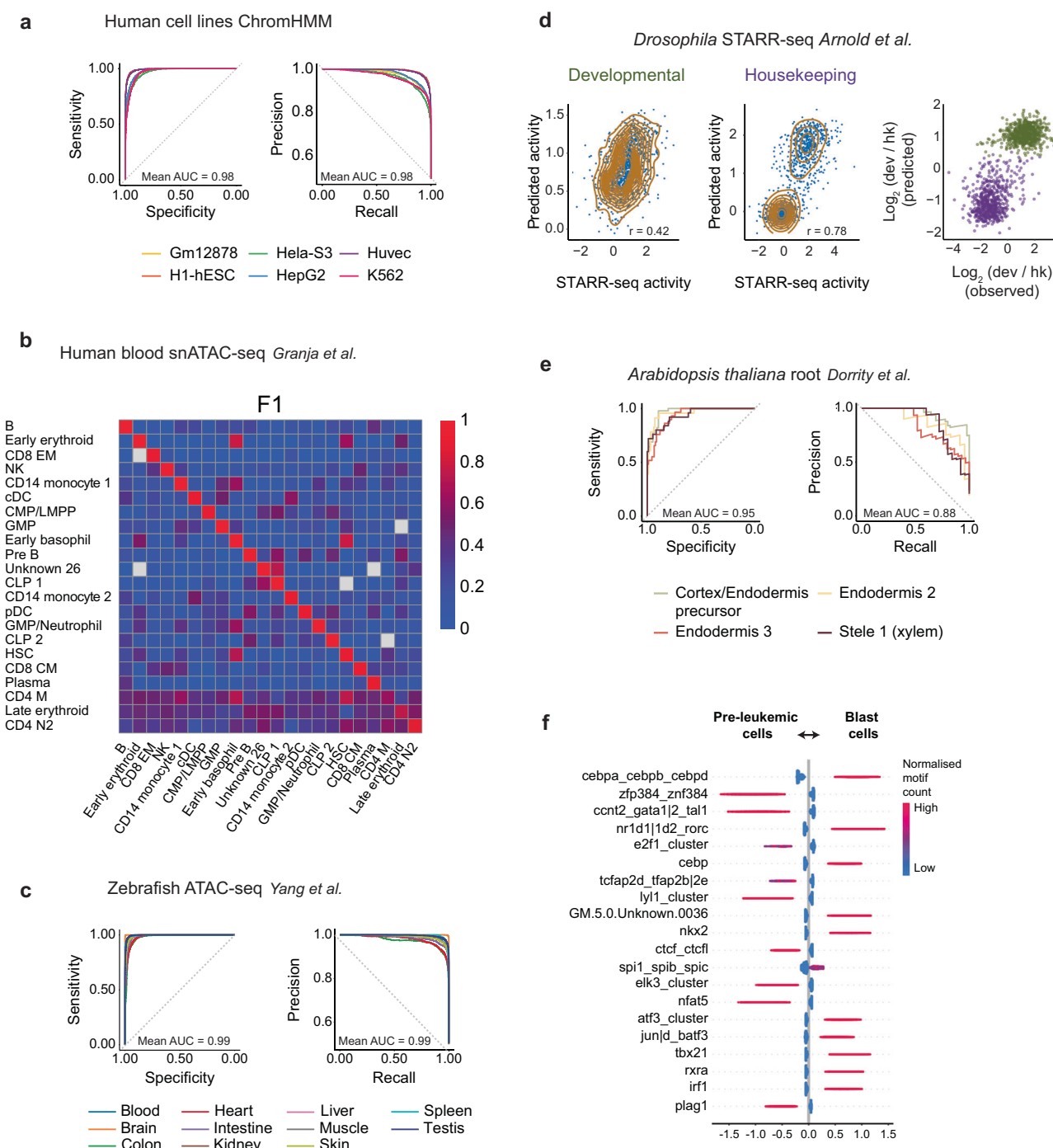

**Fig. 3 | BOM accurately classifies context-specific CREs in different datasets.**
**a** ROC curves (left) and precision-recall curves (right) illustrating the performance of binary BOM models in predicting cell line-specific CREs across six human cell lines (Gm12878, H1-hESC, HeLa-S3, HepG2, Huvec, K562) ($n = 66863$ CREs). The cell line-specific CREs were defined via a 25-state ChromHMM model[38]. **b** Performance of binary BOM models trained to distinguish cell-type-specific CREs for 22 human blood and bone marrow cell types[91]. Models were trained to distinguish cell-type-specific CREs from a background of CREs specific to other cell types. F1 scores were computed for each binary model (rows) and dataset (columns) ($n = 5124$ CREs). **c** ROC curves (left) and precision–recall curves (right) showing the prediction of tissue-specific CREs defined for 11 adult zebrafish tissues via bulk ATAC-seq data[99]. A binary BOM model was trained for every tissue ($n = 59553$). The mean AUC is shown in each panel. **d** Correlation between mean developmental (Dev) and housekeeping (Hk) enhancer activity, as measured by MPRA in fruit fly S2 cells, and

predicted activity (left and middle panels) ($n = 1258, 1258$; Dev and Hk enhancers)[30,39]. The log₂-fold change in Dev versus Hk enhancers for the measured activities on the MPRA and the predicted values (right panel). Enhancers are colored based on the observed class. **e** ROC curves (left) and PR curves (right) for the classification of cell-type-specific CREs of four *A. thaliana* root cell types from ref. 40 in a multiclass BOM model. Mean area under the curve values is shown in each panel. **f** The 20 most predictive motifs in a binary model classifying peaks more accessible in pre-leukemic or blast cells is shownn. Each dot is a single CRE. Y-axis label is a TF motif from GimmeMotifs. Color code represents the normalized motif count: (counts − min(counts, na.rm = TRUE))/(max(counts, na.rm = TRUE) − min(counts, na.rm = TRUE)). A positive SHAP score indicates importance in AML, while a negative value indicates importance for pre-leukemic. Source Data are provided as a Source Data file.

highlighted AP-1 (JunD) and CEBP motifs among the top predictive features (Fig. 3f). Such motifs form composite binding sites that have been shown to bind JUN−CEBP heterodimers and are enriched in AML-specific hypersensitive regions[42,43], consistent with the model's emphasis on these factors.

Overall, BOM maintained high accuracy across species, cell types and disease states, demonstrating that a simple motif-count representation can capture context-specific cis-regulatory grammar across a broad phylogenetic range.

## Cross-species predictions identify constraints beyond sequence alignment

Enhancers exhibit a rapid rate of evolution. In a study of human liver candidate enhancers, only 1% displayed strong sequence conservation among placental mammals[44]. This high evolutionary rate is advantageous in terms of mutational robustness, as the flexible nature of regulatory sequences allows them to maintain function despite significant sequence divergence[45]. This flexibility, however, makes their identification using traditional sequence alignment methods challenging[23,46,47]. Sequence conservation, long used as the foundation for annotating functional sequences across species, is inadequate for the identification of most mammalian enhancers[44,48–51]. Modern machine-learning approaches, such as the Tissue-Aware Conservation Inference Toolkit (TACIT)[52], bypass this limitation by training convolutional neural networks on enhancer activity and predicting conservation of open chromatin rather than nucleotide identity. To explore this idea from a motif perspective, we investigated whether BOM can capture generalizable sequence features between cell types in cross-species comparisons.

We trained BOM models on mouse embryonic cardiomyocyte and erythroid CREs and used them to classify homologous human fetal datasets (Fig. 4a, b). The mouse models correctly identified 64.5% of human cardiomyocyte and 62.9 % of human erythroblast CREs (Supplementary Fig. 11a–c; Supplementary Tables 18, 19). Strikingly, most correctly predicted elements lacked a detectable alignment between the mouse and human genomes (UCSC liftOver, minMatch = 0.95; 90.1% and 96.6%, respectively). Even after relaxing the alignment requirement to ≥60% sequence identity, only ~30% of the correctly classified CREs could be aligned (29.9% and 37.5%; Fig. 4c; Supplementary Fig. 11d, e). Reciprocally, human-trained models performed similarly on mouse CREs (Supplementary Fig. 12). These results indicate that, although enhancer sequences diverge extensively, the combination of motifs that encodes cell-type specificity remains conserved and can be captured by BOM (Supplementary Fig. 13). Thus, motif-based models detect functional conservation beyond conventional sequence alignment.

To test whether these patterns generalize across cell types, we trained multiclass BOM models on seven shared adult heart cell types and performed cross-species prediction. The models achieved mean auROCs of 0.90 on human CREs and 0.85 on mouse CREs (Fig. 4d; Supplementary Tables 20–22). We extended this analysis to single-cell resolution by aggregating SHAP scores across motifs to match each human cell to the most likely mouse cell type (Methods). Considering only human cells with at least 20 cell-type-specific open chromatin regions, 83% (46,223/55,834) were correctly assigned; ventricular cardiomyocytes showed the highest concordance at 89% (Fig. 4e; Supplementary Fig. 14; Supplementary Table 23). Cell types absent from the mouse data, such as adipocytes, were predictably misclassified.

These findings demonstrate that BOM learns a motif vocabulary enabling accurate cross-species prediction of cell-type-specific regulatory elements even when the underlying sequences cannot be aligned. Such motif-centric approaches complement other sequence-based conservation metrics to help reveal regulatory constraints that shape enhancer evolution.

## BOM identifies the key motifs driving model prediction

Understanding the sequence features that govern cell identity requires not only accurate classification of CREs but also interpretable models. To this end, we employed SHAP (Shapley additive explanations), a cooperative game-theory–based method that decomposes complex models into feature contributions[53]. For each tree-based model in BOM, SHAP calculates how much each motif reduces the training loss when it is used to split the tree, providing both global (across all CREs) and local (per sequence) importance scores.

We demonstrate this interpretability on three distal elements upstream of the cardiac regulator Nkx2-5 in mouse embryonic cardiomyocytes. The dominant motifs included MEF2 and SRF, transcription factors essential for heart development – MEF2 is a core cardiac transcription factor and SRF orchestrates multiple stages of cardiac development. Their contributions varied between the three sequences, suggesting different cooperative interactions (Fig. 5a–c). Across the dataset, ~83.4% of bases within 500 bp CREs were annotated by motifs with positive SHAP values, and each sequence had on average 32 motifs with non-zero contributions.

Where matched gene expression data were available, we linked motif importance to transcription factor expression. Using single-cell RNA-seq data from mouse gastrulation[54], we aggregated SHAP values per motif and mapped them to their corresponding factors via the Cis-BP database[55]. Motifs with high importance typically corresponded to lineage-specific transcription factors (Fig. 5d–h): cardiomyocytes showed strong contributions from MEF2C, NKX2-5 and GATA factors, whereas endothelial CREs were dominated by FLI1, JUN/FOS family members and SOX7/ETV6; neural crest motifs highlighted TFAP2A. These concordances demonstrate that BOM can identify candidate driver genes underlying cell-type-specific regulatory codes.

In summary, BOM coupled with SHAP not only predicts the cell context of a given sequence but also highlights the specific motifs that influence the prediction, offering mechanistic insights into how cells establish and maintain their identities.

## BOM motifs direct human cell-type-specific expression using synthetic regulatory elements

To assess whether the motifs identified by BOM can confer cell-type specificity, we designed synthetic regulatory elements (SREs) by inserting motif sets derived from two human cell lines, HepG2 (liver) and GM12878 (lymphoblastoid), into a enhancer template[20,56]. These lines originate from different germ layers. For each cell line we selected the five motifs with the highest global SHAP importance (Fig. 6a, Supplementary Datasets 1–3). The HepG2 motif set included hepatocyte-defining factors such as HNF4A and FOXA, consistent with the observation that a small network of core transcription factors (HNF4A, FOXA, NR1H4) underlies liver identity[57]. The GM12878 set included immune-lineage motifs.

We then generated five SREs per cell type by inserting the corresponding motifs at random positions and in random order into a common muscle enhancer sequence (Fig. 6a). Each SRE therefore contained the same motif repertoire but differed in motif spacing and orientation. The activity of each SRE was measured using a minimal promoter reporter assay in both HepG2 and GM12878 cells and normalized to transfection efficiency and the basal activity of the template enhancer.

SREs containing HepG2 motifs consistently exhibited higher reporter expression in HepG2 cells than in GM12878 cells, whereas SREs with GM12878 motifs showed the opposite pattern (Fig. 6b). The basal template alone had low activity in both lines. Moreover, HepG2-derived SREs generally drove stronger expression than GM12878-derived SREs, reflecting the predominant role of a small number of core TFs in liver gene regulation[57]. Notably, expression levels varied widely among SREs carrying the same motif set. Within cell-type-specific SREs, reporter activity differed by 3- to 83-fold,

## Cross-species prediction of human fetal cardiomyocyte CREs

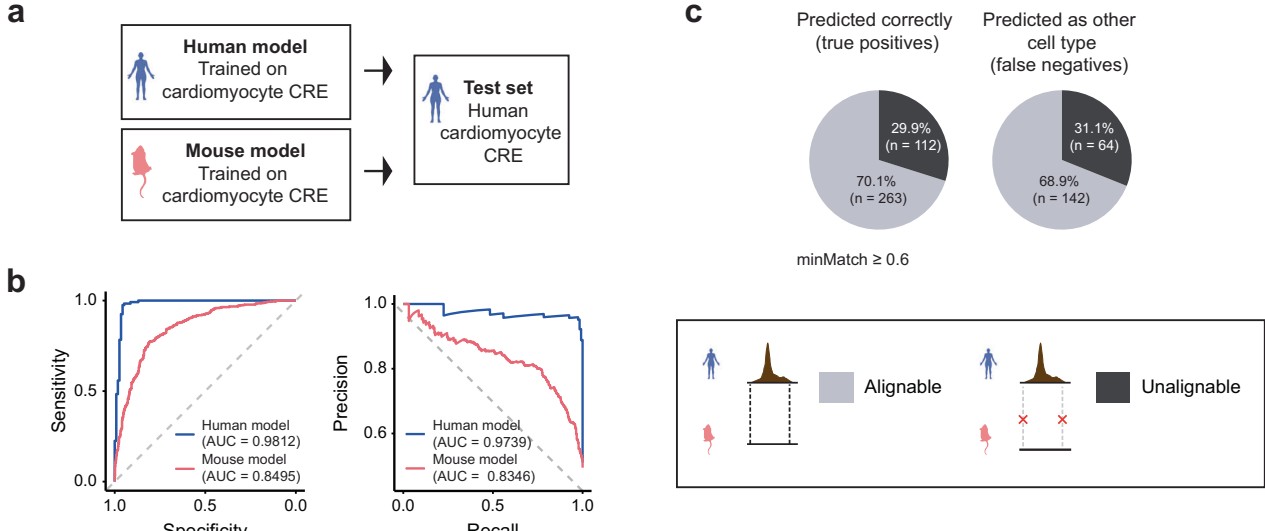

## Cross-species prediction of human adult heart cells

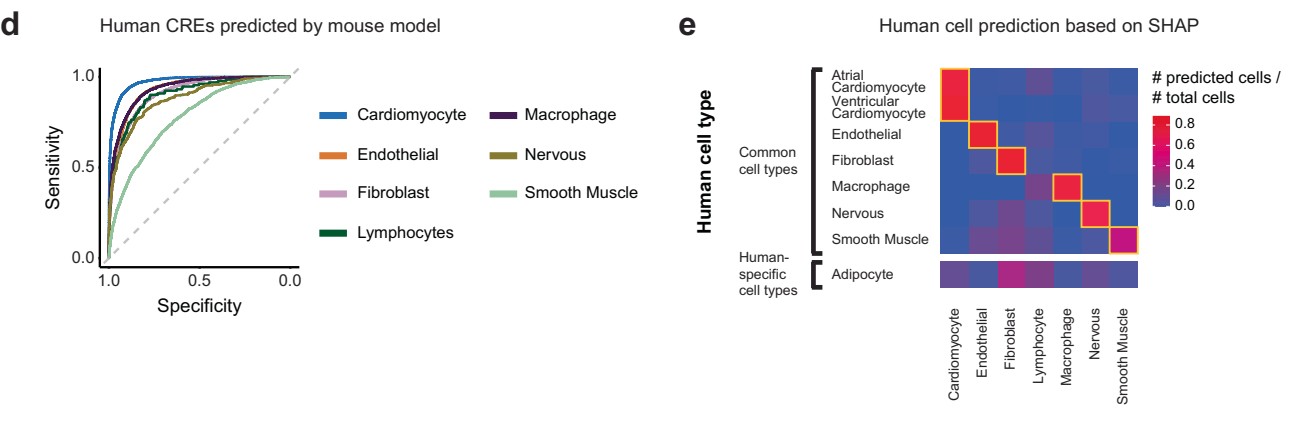

**Fig. 4 | BOM identifies cell-type-specific CREs and cell types across species.**
**a** Binary models were trained to distinguish cardiomyocyte-specific human fetal CREs or mouse E8.25 CREs from a background set of CREs specific to other cell types. Both models were used to predict human CREs, an independent CRE set not used during training. **b** ROC (top) and precision–recall (bottom) curves for the prediction of human fetal cardiomyocyte CREs via the models trained on human (blue) or mouse motif counts (pink). The area under the curve (AUC) value is shown for each patient. **c** Proportion of human cardiomyocyte CREs that can be aligned (gray) and cannot be aligned (black) with the mouse genome. The CREs were separated into those correctly predicted (true positives) and those incorrectly

predicted as the background set (false negatives). The CREs were mapped to the mouse genome via liftOver with a threshold of minMatch ≥0.6, where minMatch represents the minimum ratio of bases that should remap. **d** ROC curves for the prediction of human adult heart CREs via a mouse multiclass model. **e** Proportion of human cells ($n = 55,886$ cells) predicted as each of the cell types in the mouse multiclass model. The top section of the heatmap shows the common cell types between humans and mice, and the cell types specific to the human dataset (i.e., adipocytes) are shown at the bottom. Intersections between the human cell types and the mouse model classes are highlighted in yellow squares. Source Data are provided as a Source Data file.

indicating that motif arrangement strongly influences enhancer strength. Similar findings have been reported in systematic muta-genesis studies, where synthetic enhancers containing identical motifs (e.g., two GATA and two AP-1 motifs) showed activities spanning two orders of magnitude depending on the spacing and order of those motifs[58–60].

In summary, these results align with the concept that the identity of TF binding motifs dictates cell-type specificity, while the precise arrangement of those motifs, i.e. the orientation, spacing and order of binding sites, fine-tunes expression levels. Our experiments indicate

that BOM provides a robust strategy for assessing cell-type-specificity in the context of enhancer design.

## Discussion

DNA sequences encode regulatory information in complex and flexible ways. Despite comprehensive genome sequences for various mam-mals and detailed maps of genomic features, the challenge of under-standing distal elements that modulate spatiotemporal gene expression remains. We evaluated whether cell state can be quantita-tively captured based on distal DNA sequences alone, analogous to

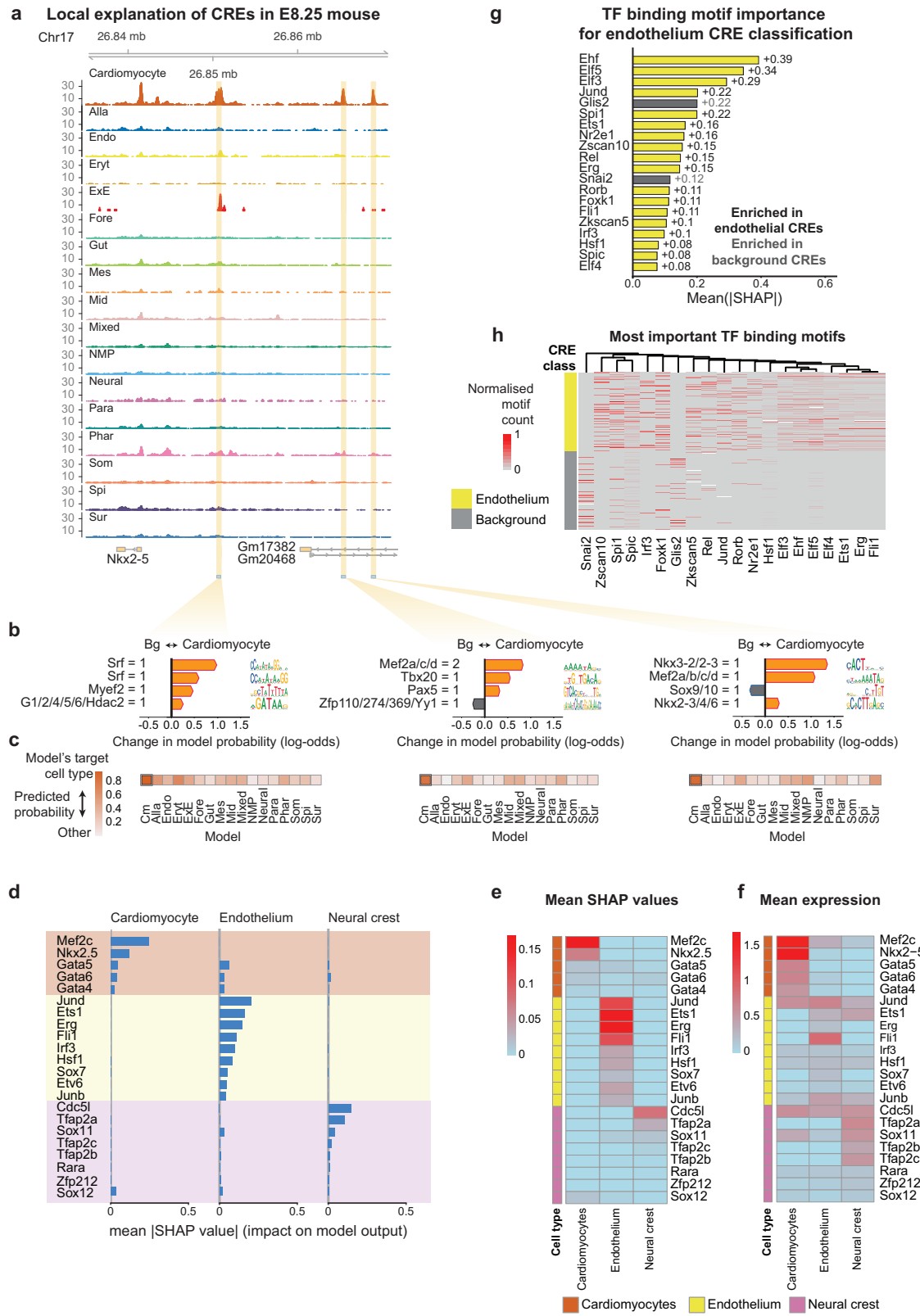

asking whether cis-regulatory sequences contain sufficient sequence information to predict the cell type where it exhibits biochemical activity. To address this, we developed a 'bag-of-motif' approach for classifying and interpreting the cis-regulatory sequences of different cell states. Focusing on distal elements, known for their context dependency, we show that the count and combination of TF binding motifs are remarkably predictive of cell states in both embryonic and

adult vertebrates. The model also performs well in modeling intermediary cell stages during embryonic development.

Motif-based annotation of regulatory sequences is not new. Unlike many motif-based tools that test for the statistical enrichment of motifs, BOM calculates predictive ability, assessing whether the sequence features are sufficient to generalize to unseen cases. Most approaches also lack local interpretability and do not identify which

**Fig. 5 | BOM provides local and global motif importance scores. a** Genome browser tracks showing the snATAC-seq signal around a region of mouse chromosome 17 near the Nkx2-5 gene for each mouse E8.25 cell type. Locations of the three cardiomyocyte-specific CREs are shown at the bottom. **b** SHAP local explanation of three cardiomyocyte-specific CREs shown in (**a**). The top four most important motifs for classifying those CREs are shown. The red and blue arrows indicate the sign (and direction) of the SHAP values. A representative name was given to every motif, and the motif count is indicated. **c** Heatmaps representing the predicted probability of the CREs shown in (**a** and **b**) by each of the binary models trained to predict CREs specific to a cell type in mouse E8.25. **d** Mean |SHAP values| of the top TF binding motifs in distinguishing cardiomyocyte-, endothelium-, and neural crest-associated CREs in mouse E8.25. The SHAP values were calculated based on the sets of CREs specific to other cell types. **e** Mean SHAP values for the TF

binding motifs are shown in (**d**). **f** Mean expression of the TFs that bind to the motifs in (**d**) and (**e**). Expression data are normalized counts from matched scRNA-seq experiments[54]. **g** Top 20 most important motifs in distinguishing mouse E8.25 endothelium CREs. Motifs are ranked by the mean of the absolute SHAP values. Bars colored yellow or gray, represent positive and negative SHAP values, respectively. **h** Normalized motif counts of the motifs shown in (**g**). The motif counts normalized as (**d**). The CREs shown are from the dataset used for endothelium CRE classification, where the background set is composed of CREs specific to the other 16 cell types in mouse E8.25 ($n = 1760, 1768$; background and endothelium CREs, respectively). Motifs were hierarchically clustered. Cis-BP 2.0 motifs for *M. musculus* from the MEME Suite were used in this figure for direct comparisons between motifs and TFs[55]. Source Data are provided as a Source Data file.

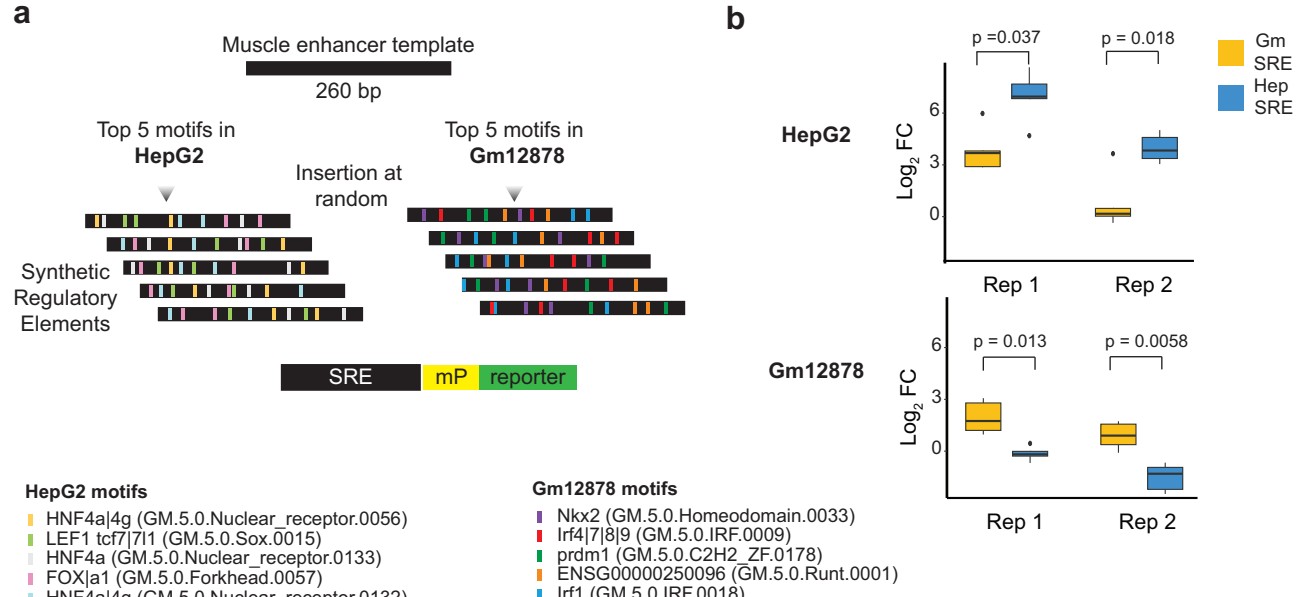

**Fig. 6 | BOM identifies motifs that drive cell type-dependent activity. a** The top 5 ranked SHAP motifs for HepG2 and Gm12878 cells were implanted into a common template sequence (2 copies of the motif in each sequence) to construct synthetic enhancers (SREs) (HepG2 SRE $n = 5$, Gm12878 SRE $n = 5$). Each SRE was inserted into a reporter containing a minimal TATA promoter and transfected into both cell types. **b** Boxplot showing the log2-fold change in the number of SREs accounting for the transfection efficiency and template enhancer activity. Two replicates of the

assays were performed. *p*-values from two-tailed t tests are shown. The quartiles in the box plots represent the 25th, 50th (median line), and 75th percentiles. The interquartile range (IQR) is the difference between the 75th and 25th percentiles. The upper and lower whiskers extend to the maximum and minimum values within 1.5 IQR above the 75th percentile and below the 25th percentile, respectively. Source Data are provided as a Source Data file.

specific motifs contribute most to any given sequence (Supplementary Table 1). BOM explains each sequence based on the set of the most relevant TF motifs to that specific prediction. This helps to define the collectives of motifs that shape different cell states. We further demonstrate that BOM can classify cell type-specific regulatory elements that do not share genome-level alignment between humans and mice, illustrating the conservation of cell type-specific regulatory grammar across species, which has also been detected using deep learning models[52,61–63].

In recent years, deep learning models have been used to learn regulatory patterns in genomic DNA[45,64,65]. Due to their ability to learn complex patterns from large datasets, deep learning models trained on large-scale genome-wide experiments, e.g., thousands of human and mouse epigenetic and transcriptional profiles across hundreds of cell types/lines and time points[45], have been used to infer regulatory signals from DNA sequences[15,17,33,66–69]. Applications range from predicting the impact of variant perturbations on TF binding, chromatin state, and chromatin accessibility (e.g., DeepSea[70], Sei[68], Basenji[71], BPNet[15]) to the sequence vocabulary of transcriptional initiation sites[66], as well as inferring evolutionary conservation of chromatin

accessibility[52,72]. Recent methods have also focused on inferring how changes in the genomic sequence alter gene expression (e.g., Enformer[17], Borzoi[69]). K-mer-based feature enumeration has been a popular method for sequence classification and variant effect prediction by combining support vector machines (SVMs) with gapped k-mer kernel functions in gkmSVM and LS-GKM[11,12,73–76].

While deep neural networks are powerful tools for genetic research, we showed they can struggle to classify cell-type-specific regulatory elements at distal regions accurately. BOM outperformed the foundational deep learning models, DNABert and Enformer, on cell-type classification. We made modifications to Enformer to enhance its ability for our cell-type classification. We introduced an adaptive layer, consisting of two fully connected linear layers tailored to the new data and cell-type contexts. We adjusted the cropping to focus precisely on the CRE of interest, thereby reducing noise to enhance the model's precision. We also performed extensive exploration of the hyperparameter space to improve Enformer's accuracy. We suggest that the performance of sequence-based deep learning methods is likely linked to the limited amount of data available to train the large number of cell states[18]. Based on our results,

transfer learning on pretrained deep neural networks (DNNs) is a promising avenue to developing accurate representations of the regulatory contributions at distal elements. The multispecies datasets used here may offer a valuable resource for testing new models toward that goal.

There are caveats to our study. While we know that active cell-type-specific regulatory regions are highly motif dependent, we have yet to determine how much of the remaining genome regions may share similar motif composition yet do not share regulatory activity. However, we demonstrated that BOM is highly specific at recognizing the distinction between flanking CRE regions and the actual CREs themselves. The performance of BOM trained at cell-type-specific regions decreases at pleiotropic distal regulatory regions. This is perhaps unsurprising as pleiotropic elements are likely to possess context-dependent accessibility, as demonstrated between sequence preferences at housekeeping versus developmental enhancers in *Drosophila*[30], which BOM was not trained to detect. In addition, BOM is constrained to cis-regulatory information within the input sequence. The status of open chromatin, histone marks, and enhancer activity can be influenced by regions beyond the sequence length used in BOM. In addition, the reliance on known TF motifs neglects informative sequences not captured by current motif annotations.

Furthermore, TFs do not bind all their motifs[5]. Preferences for certain genome domains or nucleosome regions may restrict which motifs are bound and which regions are accessible[77-79]. In regions of the genome less accessible to TFs, TF concentration may vary, which may lead to lower affinity of binding, which could particularly impact regulatory elements that are more dosage-sensitive[80]. This could influence the types of CREs identified and impact BOM predictions. However, methods that measure open chromatin appear to be generally good predictors of in vivo binding for most TFs[4]. We also know that besides direct TF binding sites, nearby sequences can impact TF binding and context-dependent enhancer activity[81-87], motifs influencing DNA shape are also enriched in regions bound by TFs[85]. These sequence rules may be better captured using other strategies, such as k-mer enumeration[12,13] or DNNs[14,15,66].

In conclusion, BOM provides a flexible, interpretable framework for deciphering cell-state-specific regulatory activity from sequence. By focusing on motif composition, it reveals a predictive vocabulary of distal regulatory sequences across species and highlights key motifs that drive condition-specific cell identity. While integration with motif grammar remains an important future direction, its predictive ability allows assessment of whether sequence features are sufficient to generalize, which is key not only for mechanistic understanding but also practical applications, such as guided enhancer design.

## Methods

### Genome builds and CRE definition
All analyses used reference genomes mm10 (mouse), hg38 (human adult heart from Hocker et al.[88]), and hg19 (other human datasets). For fruit fly, zebrafish, and *Arabidopsis thaliana*, dm3, danRer10, and TAIR10 builds were used. Candidate cis-regulatory elements (CREs) were defined from various assays (scATAC-seq, bulk ATAC-seq, ChIP-seq, ChromHMM, STARR-seq) and required to be >1 kb from transcription start sites and nonexonic. CRE coordinates were processed using the R packages GenomicRanges and GenomicFeatures (versions 1.42.0/3)[89]. Each CRE was trimmed or centered to a fixed window (generally 500 bp ± 250 bp from the peak summit; specific details per dataset below). Supplementary Table 2 lists the number of CREs per context.

**Mouse embryonic day 8.25.** scATAC-seq peaks from Pijuan-Sala et al.[25] were assigned to cell types by Fisher's exact tests. Peaks annotated to multiple cell types or unannotated were discarded; notochord peaks (65 distal CREs) were removed due to low counts. Each peak was

trimmed to 500 bp. An additional classifier was trained on 93 latent regulatory topics derived from cisTopic[27] ($n = 76,604$ multitopic CREs removed).

**Mouse adult heart.** Marker peaks were identified from snATAC-seq data (10.5281/zenodo.15720256) using Seurat's FindAllMarkers with avg_log2FC > 0 and adjusted $p < 0.05$ (version 4.1.0)[90]. Peaks were filtered to a single-cell type, distal (>1 kb), nonexonic, and trimmed to 500 bp from midpoint ($n = 24,841$ CREs across 16 annotated cell types plus "Unknown").

**Human cell lines.** Six ENCODE lines (Gm12878, H1-hESC, HeLa-S3, HepG2, HUVEC, K562) were processed from 25-state ChromHMM annotations[38]. Regions labeled Enh/EnhF were selected, and any enhancer overlapping strong enhancers in other lines was removed, yielding 167,148 candidate enhancers (centered at 194 bp).

**Human hematopoiesis.** scATAC-seq peaks (Granja et al.[91]) for 26 blood and bone-marrow cell types were assessed for cell specificity via Fisher's exact tests (Bonferroni-adjusted, alternative = "greater"). Peaks had to occur in ≥5% of cells and achieve adjusted $p < 0.01$ for the global test and adjusted $p < 1 \times 10^{-10}$ in up to 13 pairwise tests[25]. Peaks annotated to multiple cell types were discarded; 12,796 cell-type-specific peaks remained (500 bp each). Two clusters with <12 peaks were removed ("unknown 13" and "unknown 14").

**Human adult heart.** snATAC data from Hocker et al.[88] provided 19,447 peaks (fold-change >1.2, FDR < 0.01 via edgeR (version 3.24)[92]). After filtering for distal (>1 kb) and nonexonic regions, 16,381 CREs were retained and centered to 500 bp.

**Human fetal tissues.** Domcke et al.'s atlas (15 organs)[93] was processed similarly to the hematopoiesis pipeline. Mixed cell-type annotations ("Cardiomyocytes.Vascular_endothelial_cells") were excluded. Cell types present in multiple organs were aggregated, leaving 71 distinct cell types. Peaks with adjusted $p < 1 \times 10^{-10}$ in up to 20 pairwise tests were retained. All CREs were trimmed to 500 bp.

**Acute myeloid leukemia (AML).** ATAC-seq data for pre-leukemic hematopoietic stem cells and blast cells (Corces et al.[41]) were processed using Bowtie2, MACS3, and DiffBind. individual IDs were SU484, SU501, and SU654. Adapters were identified and removed via BBDuk (ktrim = r k = 23 mink = 11 hdist = 1 tpe tbo; http://jgi.doe.gov/data-and-tools/bb-tools/). The reads were aligned to the hg19 assembly with Bowtie2 v.2.3.5.1 in paired-end mode[94]. Poor-quality reads were discarded (-f2 -q30 -b -S), and the output was sorted with SAMtools (v.1.10)[95]. MACS3 was used to call peaks in each sample (-g hs -f BAMPE -B and default $q$ value cutoff of 0.05)[96]. Differentially accessible peaks (DAPs) were identified via DiffBind[97]. The function dba.count (with options summits = 200 and bUseSummarizeOverlaps = TRUE) was used to count reads on the summits of the narrow peaks called with MACS3. The function dba.analyze was used to identify DAPs between AML and healthy samples via DESeq2 (v.1.30.1)[98] and edgeR[92]. A total of 26051 and 45569 DAPs were identified with edgeR and DESeq2, respectively (FDR ≤ 0.05). Differentially accessible peaks with FDR ≤ 0.05 (edgeR and DESeq2) and $|\log_2 FC| > 1$ were retained (25,688 peaks).

**Zebrafish.** Adult tissue ATAC-seq peaks from 11 organs[99] were trimmed to 500 bp, yielding 135,763 CREs.

**Fruit fly.** Enhancers defined by STARR-seq in S2 cells were classified as housekeeping or developmental[30]; after excluding enhancers used to train DeepSTARR[39], 6287 sequences of variable length remained.

**A. thaliana root.** scATAC peaks were clustered by cell type[40]; cluster-specific peaks (distal ± 500 bp) were extended to ±300 bp (600 bp total) from center. Only clusters with ≥100 CREs (precursors of cortex/endodermis, endodermis 2/3, and xylem) were retained.

## BOM model

For each context (cell type, condition or topic), a motif-count matrix was assembled (rows: CREs; columns: motifs). Analyses required ≥100 unique CREs per context. For binary classification, positive CREs (from the target cell type) were balanced against equal numbers of negative CREs sampled from other cell types.

**Motif database.** CREs were annotated with vertebrate motifs from the GimmeMotifs database (v 5.0; 1,796 motifs clustered to reduce redundancy)[10] (https://github.com/vanheeringen-lab/gimmemotifs), and, for *A. thaliana*, motifs from CisBP (v 2.0). Motifs were scanned using FIMO with threshold 0.0001 (*q*-value ≤ 0.5). To assess the impact of overlapping motifs, overlapping matches were removed by keeping the highest-scoring motif per region; training on this reduced set revealed the effect of motif overlap. Motif counts per sequence formed the feature matrix for training.

**Motif scoring.** Motif instances were identified for each sequence via FIMO from the Meme Suite, with a threshold of 0.0001 (--thresh 0.0001)[26]. FIMO scores the similarity between the input sequences and the motif PWMs. FIMO incorporates a background model to determine the expected scores for random sequences lacking the motif. FIMO will score a PWM in both forward and reverse directions. Our analysis applied a *q* value cutoff of ≤0.5 to retain motif instances.

**Building the classification data matrix.** To create datasets for context-specific classification, we built a matrix of motif counts for each cell type where rows represent candidate CREs and columns represent TF binding motifs. We required a minimum of 100 unique peaks per context (cell type/condition/topic). For binary classification, every matrix contained a set of CREs of the target cell type (positive class) and a similar number of CREs from the rest of the cell types (negative class, balanced with stratified sampling to ensure equal contributions from every nontarget cell type). In cases where there are not enough CREs from a cell type to contribute to the background set, we lowered the contribution of every background cell type to that minimum number and adjusted the number of sequences from the target cell type.

**Gradient-boosted decision trees.** Models were trained using XGBoost (R package v 1.6.0.1). Data were split 60%/20%/20% into training/validation/test sets. Binary models used the "binary:logistic" objective with nrounds = 10,000, eta = 0.01, max_depth = 6, subsample = 0.5, colsample_bytree = 0.5, and early_stopping_rounds = 100. Multiclass models used objective = multi:softprob and evaluation metric "mlogloss"; multi-label classification was implemented as a set of binary models (one per class). Regression models (for enhancer activity) used objective = reg:squarederror and eval_metric = rmse. Predictions were carried out with the function "predict" in all cases (R package stats version 4.0.0). Predictions ≥0.5 were considered positive

**Model evaluation.** Performance metrics included precision, recall, F1, Matthews correlation coefficient (MCC), area under the ROC curve (auROC), and area under the precision–recall curve (auPR). For regression, Pearson correlation between predicted and measured enhancer activities was reported. ROC and PR curves were generated using cvAUC (version 1.1.4) and yardstick (version 1.1.0) in R.

**Calculating motif importance.** Shapley values[100] were used to quantify the contribution of each feature (i.e., motif) to the model's prediction for each data instance (i.e., CRE). We calculate Shapley values

via Shapley additive explanations (SHAP)[101]. We trained models with every combination of motifs as features. Once these models are trained with the same parameters and data, we predict a single observation using all of them. The difference in prediction when a motif is added represents its marginal contribution. The overall effect of a motif is then computed as the average marginal contribution across all models where the motif is included. These marginal contributions are aggregated via a weighted sum. The weights assigned to each model depend on the number of features included. The weight of a model with n features is calculated as one divided by the number of models with N features. This ensures that the sum of the weights for all the models with the same number of features equals 1. SHAP values are explained and calculated with the function explainer (python SHAP v 0.41.0). A SHAP value is calculated for every motif and every CRE. We use rank motif importance as the sum of absolute SHAP values.

**Assessing the impact of overlapping motifs on model performance.** To investigate the impact of overlapping TF binding motifs in CRE sequences, we focused on mouse E8.25 CREs and generated a set of nonoverlapping motifs for every cell type. To filter the motifs, we ordered them on the basis of their start positions and systematically removed overlapping motifs with lower match scores until no motifs overlapped. When two overlapping motifs had equally high scores, one was selected at random. Following removal, there was an average of 42.6 motifs per 1 kb (*q* value ≤ 0.5). We subsequently constructed motif count matrices and trained models using the abovementioned parameters.

**Multiclass models of adult hearts.** Using human[88] and mouse datasets[102], we trained models to classify CREs among seven common cell types (cardiomyocyte, endothelial, fibroblast, lymphocyte, macrophage, nervous cells, and smooth muscle). To train the seven-class models, CREs from similar cell types were aggregated: human atrial and ventricular cardiomyocyte CREs were aggregated into the "cardiomyocyte" class; mouse coronary, endocardial, and lymphatic endothelial cell CREs were aggregated into the "endothelial" class; B cell, T cell, and NK cell-specific CREs were aggregated into the "lymphocyte" class; and CREs of the M1 and M2 macrophage types were aggregated into the "macrophage" class.

**Negative flanking set.** Negative flanking sets were constructed by sliding 500-bp windows (50-bp stride) ±2 kb from each mouse E8.25 CRE, excluding overlapping windows; the trained binary models were applied to these flanking sequences to estimate false positive rates.

**Super-enhancer analyses.** Super-enhancers were identified with ROSE (Rank Ordering of Super-Enhancers)[103] (stitch distance 12.5 kb; ±2.5 kb TSS exclusion) using H3K27ac ChIP-seq data from E10.5 mouse heart and forebrain (Accession: ENCSR825ZJV and ENCSR58SSPN). CREs overlapping more than one peak were labeled "super-enhancer" and compared with classical CREs from the E8.25 test sets.

## Benchmarking against alternative models

We used the same input and output configurations across all the deep learning architectures. The input was represented via one-hot encoding of 500 nucleotide (nt) sequences, whereas the output layer consisted of a fully connected layer with 17 units via a sigmoid activation function. We employed identical hyperparameters, including 100 epochs, a batch size of 128, early stopping of 10, and validation loss calculated via the 'binary-crossentropy' metric, to train the models. We evaluated the best model based on loss.

To compare BOM with other approaches, LS-GKM and DNABERT models were trained on the same training/validation sets for mouse embryonic cell-type classification[12,29]. LS-GKM used default parameters (-t4 -l11 -k7 -d3) and was trained on concatenated training + validation

sets. DNABERT models ($k = 6$) were fine-tuned for each cell type; epochs (5–20) and learning rates (0.0001–0.0003) were explored, and the best model by validation loss was selected (Supplementary Fig. 15). Enformer was fine-tuned by adding a two-layer adaptive module to the pre-trained model, cropping 704 bp around each CRE, and searching hyperparameters (batch size 4–8, layer size 8–32) via Bayesian optimization using the ray library (2.7.0)[104]. The Async Successive Halving ASHA[105] scheduler uses the Bayesian Optimization HyperBand (BOHB)[106] search algorithm. Each cell type was tested with ten samples up to 10 epochs. The checkpoint with the lowest loss was then used to test the model performance on the test set.

Three CNN-based architectures (CNN×3+FC×2, CNN+LSTM, CNN×4+FC×2) were implemented for multiclass classification and trained with reverse-complement augmentation. These deep learning architectures were selected based on convolutional neural networks and hybrid architectures used by DeepMEL[31], DanQ[32], DeepSTARR[30], and Basset[33]: CNNx4+FCx2, CNN+LSTM, and CNNx4+FCx2 CNNx3+FCx2, respectively (Supplementary Fig. 16). Hyperparameters (learning rate $1 \times 10^{-5}$–$1 \times 10^{-1}$, batch size 64 or 128) were explored via a Bayesian hyperband search.

Basset and DeepMEL pre-trained weights were fine-tuned with frozen convolutional layers. The ray library (2.7.0)[104] was used for a hyperparameter grid search. The learning rates for each model were sampled from a uniform distribution, ranging from $1e^{-5}$ to $1e^{-1}$, and the batch sizes were evaluated at 64 and 128. Each model was tested with 40 samples up to 30 epochs. Checkpoint with the lowest loss was then used to test the model performance on the test set (Supplementary Table 25).

To fine-tune Basset, we reimplemented the Basset models using published hyperparameters. We configured the training process with 60 epochs with an early stopping of five epochs. We froze the pre-trained weights and fine-tuned the final output layer to tailor the model's predictions to the 17 distinct cell types. The ray library (2.7.0)[104] was used for a hyperparameter grid search. The learning rates for each model were methodically sampled from a uniform distribution, ranging from $1e^{-5}$ to $1e^{-1}$, and the batch sizes were evaluated at 64 and 128. The BOHB[106] search algorithm was used. Each model was tested with 40 samples up to 30 epochs. Checkpoint with the lowest loss was then used to test the model performance on the test set (Supplementary Table 25). A grid search on the original architecture was performed for DeepSTARR. We did not fine-tune owing to species divergence between insects and vertebrates.

## Cross-species analyses
Binary models trained on mouse E8.25 cardiomyocyte and erythroid CREs were applied to human fetal cardiomyocyte and erythroblast CREs and vice versa. CREs were lifted over between hg19 and mm10 using liftOver (v 435) with minMatch = 0.95 or 0.6.

**Cross-species cell alignment.** We scored mouse and human marker peaks using the mouse multiclass model and aggregated mean SHAP values across all accessible peaks for each cell, motif and model class. Accessibility was defined as count > 0 in the mouse snATAC-seq matrix and exactly 1 in the human binary snATAC-seq matrix. Only marker peaks belonging to the same cell type or cluster were considered, and for mouse cells we excluded any CREs used during model training or validation. SHAP values were normalized per cell by dividing each value by the sum of the absolute SHAPs for that cell. Integration was performed on 10,000 cells per species: if a cell type contained more than 1000 cells, it was down-sampled to 1000. Cells with fewer than 20 accessible marker peaks were excluded. The normalized SHAP matrices (cells × motifs × classes) were then integrated across species using Seurat v4.1.0[90].

**Cross-species cell prediction.** To assign each cell to the most likely class, we calculated, for each model class, a z-score-transformed ratio of the sum of positive SHAP values to the sum of absolute SHAP values.

SHAP values were normalized per cell as above. The predicted label for each cell was assigned to the class with the highest aggregated SHAP score.

## Gene expression data processing
**Mouse E8.25.** We used a single-cell RNA-seq mouse embryo time course atlas with gene expression spanning multiple developmental stages from E6.5 to E8.5[54]. Using the expression and metadata matrices, cells with fewer than 1000 expressed genes, those with a mitochondrial gene expression fraction higher than 2.37%, and those with a mitochondrial read fraction outside their upper-end distribution were excluded, resulting in 116,312 cells[90]. Expression was normalized via the function NormalizeData from Seurat (v 4.1.0)[90] with default options (normalization.method = "LogNormalize", scale. factor = 10,000).

## Activity testing of synthetic regulatory elements
We selected the top five explanatory motifs for HepG2 and Gm12878 from the ENCODE human cell line model we constructed. We used a muscle cell-specific enhancer as a sequence template (chr2: 211,153,238–211,153,405), which was previously used for a similar purpose, and extended this to 260 bp (chr2:211,153,192–211,153,452)[20,107]. We constructed 5 SREs for each cell line by inserting two copies of each motif in random order and locating them into the template to a final length of 260 bp. We tested the 10 SRE and the template for enhancer activity in both cell lines via a luciferase assay with a minimal promoter. Gm12878 cells (Corielle Institute of Medical Research) were cultured in RPMI media supplemented with 15% FBS and 2 mM L-glutamine. HepG2 cells were a gift from N. Turner (VCCRI) and were cultured in DMEM containing 10% FBS.

qPCR-based validation of the synthetic enhancer was performed as previously described[108]. Briefly, the fragment to be tested was inserted upstream of the minimal TATA box promoter of the pGL4.10 (luc2) vector (Promega #E6651). Five million cells were then electroporated with the vector along with 2 µg of a Renilla luciferase (RLuc) control plasmid (Promega #E2251). After electroporation, RNA was isolated after 6 h, treated with DNase, and reverse transcribed using the primers shown below. qPCR was performed, and the results were analyzed via the delta–delta Ct method to quantify the expression levels, accounting for the transfection efficiency and activity of the muscle enhancer template in each cell type. Three technical replicates of qPCR were averaged. Two replications of the enhancer assays were performed using separate batches of cells.

FLuc F: CATTAAGAAGGGCCCAGC; FLuc R: GCTTCATAGCT TCTGCCAG;

RLuc F: TAACTGGTCCGCAGTGGTG; RLuc R: TGGCACAACATGTC GCCA

## Reporting summary
Further information on research design is available in the Nature Portfolio Reporting Summary linked to this article.

# Data availability
Processed datasets used in this paper are available at Zenodo (https://zenodo.org/records/10280300). Mouse E8.25 scATAC-seq from GEO accession GSE133244[25]. Mouse E8.5 data from GSE205117[28]. Mouse scRNA-seq embryonic stages E6.5 to E8.5 from E-MTAB-6967[54]. Mouse adult heart data was obtained from ref. 109. ChromHMM models for human cell lines (Gm12878, H1-hESC, HeLa-S3, HepG2, HUVEC, and K562) were downloaded from the UCSC Genome Browser (https://genome.ucsc.edu/cgi-bin/hgFileUi?db= hg19&g=wgEncodeAwgSegmentation)[38]. Human hematopoiesis chromatin accessibility data were obtained from (https://jeffgranja. s3.amazonaws.com/MPAL-10x/Supplementary_Data/Healthy-Data/ scATAC-Healthy-Hematopoiesis-191120.rds)[91]. Human adult heart

data comes from Hocker et al.[88]. Human fetal chromatin accessibility data was obtained from GSE149683[93]. ATAC-seq files for the matched pre-leukemic and blast cells were downloaded from GSE74912 (SU484, SU501, and SU654)[41]. Fruit fly S2 cells STARR-seq enhancer activity data were obtained from GSE40739 and GSE57876[39]. Zebrafish ATAC-seq data was obtained from GSE134055[99]. Arabidopsis thaliana root CREs were obtained from GSE173834[40]. Source data are provided with this paper.

## Code availability

The code used to develop the model is available as R package with tutorial at https://github.com/ewonglab/BOM_package/. The code used to perform the analyses for the manuscript are available at https://github.com/ewonglab/BOM_manuscript_data_scripts[109]. Information on running times can be found in Supplementary Fig. 17.

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

## Acknowledgements

We kindly thank D. Humphreys (VCCRI), X. Guo (VCCRI), and M. Niu (VCCRI) for computational advice and support. We also thank F. Zanini (UNSW), M. Francois (USyd), and members of the Wong Lab for providing feedback. We thank the Victor Chang Innovation Centre for its support. This project was undertaken with assistance from the resources and services of the National Computational Infrastructure (NCI), supported by the Australian Government. We are grateful for the support of the UNSW NCI Resource Allocation Scheme (https://doi.org/10.26190/PMN5-7J50), NHMRC Investigator Grant (GNT2009309), ARC Discovery Project (DP200100250), Snow Medical Fellowship to E.W.

## Author contributions

P.C., X.Z., Z.L. performed the computational experiments, data analyses, and code development. L.L. and Y.Y. performed the experimental assays and data analyses. P.C. and X.Z. contributed to manuscript preparation. E.W. conceived the study, designed the experiments, analyzed the data, and led the manuscript preparation. All the authors reviewed and approved the final version of the manuscript.

## Competing interests

The authors declare no completing interests.
