## [Transparent Peer Review file · Nature Communications]

Motif-based models accurately predict cell type-specific distal regulatory elements

Corresponding Author: Dr Emily Wong

Version 0:

Reviewer comments:

Reviewer #1

(Remarks to the Author)

Here, Cornejo-Paramo et al present Bag-of-motifs (BOM), a novel computational framework for learning cell-type specific regulatory activity of distal enhancer sequences. Decoding regulatory elements and identifying TF binding motifs and their impact on gene expression remains one of the major challenges in computational biology. Today, there are large volumes of data available profiling open chromatin, TF binding, and gene expression at single cell resolution. Yet, there is still a need for improved computational methods as existing ones, including deep learning based ones, have so far not become widely used towards identification of cell type specific regulators.

In light of these challenges, I found the current manuscript a welcome contribution. The accuracy achieved by their classifier compared to state-of-the-art methods is highly impressive, and BOM has the potential to help researchers pinpoint key motifs when dissecting distal enhancers. Nevertheless, I believe that the following major issues must be addressed before the manuscript can be accepted for publication:

Overlapping TF binding motifs. Why do overlapping TF binding motifs need to be removed as outlined in the methods? It is assumed that the higher-affinity motifs are more likely to bind. Is this supported by any literature? The authors should justify their approach more. Given their note on relaxed motif detection thresholds (L779), this part of the algorithm looks like a step in the opposite direction. Also, how stable is the outlined approach? For example, if the sequence is scanned from the end instead of the beginning of the CRE, how different are the results?

CRE numbers. There is a substantial variation in CRE numbers (Table S1). Is this a technical artifact, or are there some biological reasons for the variation observed? Is having fewer CREs degrading the performance of the model?

Negative controls. I would like to see additional negative control experiments to ensure that BOM picks up biologically meaningful experiments. That is, how would the model perform if it were assigned the sequences 2 kb upstream of each CRE instead of the sequence overlapping the CRE?

I understand that there are space constraints, but I was wondering if Fig 5c could be redesigned to use bars instead of a heatmap? As it stands I find it very difficult to compare the three models.

Maybe my colorvision is not sharp enough, but for Fig 5d I can only distinguish two colors, corresponding to the two extremes of the scale. Is it the case that there are no real intermediate values? If so, could the authors consider displaying the information differently? Also, are these supposed to be violin plots? I do not understand what the blue boxes and red lines represent.

Testing the method with the provided example in the tutorial.md was not easy. We had two people testing it with different levels of experience. First, the fimo command line in the tutorial gave different output, and results ended up without genomic coordinates. Running it through R worked okay with the sample input. However the output was in ensembl whereas in the tutorial is Gencode. Therefore, we could not reproduce the same output. In general, mixing of ensembl and gencode annotations is confusing, although it does not make a difference in practice. Ideally, the method should clearly outline how to deal with bed files originating from Ensembl or gencode annotations. This will facilitate the wider adoption of the tool with non-computational people.

Also, the following minor issues need to be addressed:

L83: For the multiclass model, could the authors also provide a confusion matrix to visualize what misclassifications are the most common?

L85: The authors mention a latent space embedding, but I could not find any information in the methods about how this was done. Could the authors please clarify?

L170: "sequences could not be unaligned" What does it mean to unalign a sequence (is this even a real verb)? I am not familiar with this procedure, so please clarify.

L390: Filtering-based Motif Similarity. Not enough information is provided on how this filtering happened. How is each motif represented? As PWM? How were different length motifs compared?

L292: I could not source reference 50 of the mouse adult heart.

L403: No investigation of overlapping binding motifs occurred. The authors should correct the sentence.

(Remarks on code availability)

Please see above

Reviewer #2

(Remarks to the Author)

This article presents a machine learning method, Bag-of-Motifs (BOM), for learning combinations of motifs that identify cell type specific enhancers. The method is plausible (if fairly straightforward) and shows success on a large and diverse collection of problems to which they apply it. Moreover, they perform wetlab work to confirm some predictions of functional enhancers. The main downside of the paper is that they fail to recognize much prior literature and some prior methods relevant to their work - presenting their contributions as if they are much more novel than they really are, and potentially not comparing their method to other methods that might also perform well. Although there is some novelty and considerable effort represented in the work. In more detail, the strengths and weaknesses of the article are as follows.

1) The basic algorithm they propose is that they take as input single-cell or single-nucleus ATAC-seq data, from which cell types are identified, and differentially accessible chromatin regions are identified for those cell types. They filter out regions that are too near genes. They count TF motifs in those regions, create a vector representation for each region. Then they use XGBoost to predict the cell type specific regions, and SHAP to evaluate the importance of particular motifs. This all makes sense, algorithmically and biologically.

2) However, the authors really ought to cite more prior work to show that these ideas are not coming from nowhere. For instance, representing promoters (and to a lesser extent enhancers) by vectors of motif counts has been done for decades - ever since the early post-human-genome days. There's dozens of such papers at least - too many for me to list. But the point is that representing the potential regulatory- or condition-specific character of genomic regions by vectors of motif hits has a long history. The authors should make more of an effort to find such works, cite them, and possibly even compare those methods to theirs. (Applying this idea to sc/sn-ATAC-seq regions was not done back then, of course, because such data only exists in recent years!) Indeed, the four main components of their algorithm -- differential accessibility, TF count vector representation, XGBoost, and SHAP -- are all individually established in genomics, including in the study of enhancers. The authors need to cite more of that prior work, even if/though the particular way that they have combined those ideas in the present paper has not been done before.

3) Another strength of the paper is its successful demonstration on a large number of diverse examples. They look at cell types in mouse embryonic development, human cell lines, zebra fish, human blood, drosophila, arabidopsis, cross-species analysis of cardiomyocytes, etc. Personally I would have liked to see Supp fig 4 about AML motifs in a main figure. But this is a small point. The authors have done considerable work to acquire these datasets, run their approach and some competing approaches, and intelligently analyze/interpret their results.

4) A considerable strength is also the effort to move beyond the drylab to do wetlab experiments. The authors demonstrate that they can add to an enhancer cell type specific TF motifs that they detected, and that these drive gene expression in a cell type specific manner. While synthesizing and testing enhancers in vivo is fairly straightforward for appropriately equipped wetlabs, few primarily-computational papers make any such effort (or even have the capacity or collaborators to do so). I would like to know the exact enhancer sequences they tested. I'm not sure if they're reported anywhere in the paper. (I looked but may have missed them.) They could even be potentially show in the main figure - instead of the boring grey bars, they could show exactly where the motifs were inserted in the enhancer each time. Nevertheless, this is a strength of the paper. A deeper analysis might look at the expression levels of the TFs whose motifs they inserted, and see which ones are expressed. And a yet deeper analysis might look at different combinations of TF motifs. But already the experiments represent a validation of their approach.

5) As mentioned above, a general failure of the paper is to recognize, cite, or utilize a large body of previous work on enhancer motifs, or to even recognize the state of the art in that field. Moreover, there are too many unsubstantiated claims. For instance, consider the first line of the Discussion: "Our results implies that motif composition, rather than primary sequence conservation, plays a central role in regulatory evolution, providing new insights into how gene regulatory networks adapt and diversity." Firstly, it has already been known for decades that motif composition is conserved far more than primary sequence, and evidence of this continues to roll in as ever more genomes are sequenced and compared. It is

an appalling disservice to previous work to state this as if the present paper is discovering this newly. Moreover, the present paper does not study the evolution of regulation in any substantial way. There is only one comparison of mouse versus human enhancer motifs in cardiomyocytes - far from a comprehensive evaluation of regulatory evolution. Further, the performance of their system when trained on one species and tested on another is substantially worse than when trained and tested on the same species. So it is not clear how strongly, in a quantitative performance sense, evolution is conserving motif composition in the sense that they study it. Finally, I would question precisely what 'new insights' this paper offers into regulatory network evolution. I would suggest a thorough reading of, among other things, work from the Zoonomia consortium and Andreas Pfennig and various colleagues in particular (including his TACIT algorithm), to better understand the state of the art in studying the evolution of regulatory elements, and to place the present work in more proper context.

Or, take the second line in the Discussion "Traditional methods for identifying regulatory conservation rely on sequence alignment, which often fail to detect functionally similar regulatory elements lacking strict sequence similarity." I am not sure precisely what methods they are referring to. Certainly it is normal to look for orthologous regions when studying a new species. However, it has again for decades been recognized that this is insufficient for regulatory regions. Indeed, it is often insufficient even for genes, where we know 3D structure of proteins is far more conserved than primary sequence. Thus, there is a whole collection of tools that one can use to study and try to establish links between regulatory regions in different species. (Again see Zoonomia consortium, but many others as well!) If one is talking within species, then TF motif enrichment analyses have long been one of those tools and continues to be a key approach. For instance, the MEDEA paper looks at TF motif enrichment in cell type specific differentially-accessible regions - very similar in spirit to the current paper, but not cited. And that is but one example.

6) An important but easily remedied criticism of the work is that they refer to gene regulation far too much. They study differentially accessible regions (DARs). Such regions may have a regulatory function or they may not. Some estimates suggest that many do not, and even that many H3K27ac-associated regions (which is a classic enhancer mark) are not truly regulatory but mere side effects of promiscuous marking or 3D chromatin conformation bringing together non-regulatory regions near marked regulatory regions. With the obvious exception of the final wetlab portion of their study, very little of what they study is definitively regulatory, but merely accessible, and they should change the language accordingly.

(Remarks on code availability)

I checked that the repository is there and browsed the code to some extent, but did not download and test.

Reviewer #3

(Remarks to the Author)

(Remarks on code availability)

Reviewer #4

(Remarks to the Author)

This manuscript introduces BOM (Bag-of-Motifs), a computational framework that utilizes transcription factor binding motif composition to predict cell-type-specific regulatory elements. The method employs an XGBoost model trained on motif occurrence counts and demonstrates applications across multiple species and biological contexts. While the feature attribution approach for identifying key transcription factors presents an interesting perspective, fundamental issues regarding the algorithm's innovation, biological purpose, experimental validation, and comparative analyses significantly undermine the study's conclusions. In its current form, the manuscript is not suitable for publication. The following concerns should be comprehensively addressed to support the research claims:

Major Concerns:

1. The core methodology closely resembles the established chromVAR framework in its use of motif composition for regulatory analysis. The authors must explicitly demonstrate how BOM provides meaningful improvements over chromVAR's motif accessibility deviation approach, which was not cited.
2. The study claims to predict cell-type-specific regulatory elements (CREs), but such elements are already reliably identified through standard experimental methods (e.g., differential ATAC-seq peak calling). The authors must better justify the need for computational prediction when experimental assays can directly identify these regions.
3. The manuscript suffers from a profound conceptual disconnect between its stated objectives and actual implementation. While the title and Results sections repeatedly claim that BOM predicts "cell-type-specific regulatory elements" (e.g., Result 1: "A motif-based framework...", Result 2: "BOM outperforms..."), the methodology in fact operates in the opposite direction: it uses experimentally pre-defined cell-type-specific CREs (identified through conventional differential analysis) as input to predict cell type labels.
4. The motif-based classification approach fundamentally fails to account for differential activity of transcription factor (TF) family members across cell types. While factors like BCL11A and BCL11B share similar binding motifs, they exhibit distinct regulatory roles in different immune cell differentiation lineage. The current binary motif representation cannot capture these functional differences, potentially leading to misclassification of cell states/phenotype where TF activity is modulated through post-translational modifications rather than solely binding site number.

5. The motif scanning using FIMO may introduce substantial false positives in TF binding predictions. Authors should validate their predictions by leveraging large-scale TF ChIP-seq datasets to demonstrate that identified motifs correspond to binding events in relevant cell types. Without such experimental validation, it remains unclear whether predicted motif combinations reflect genuine regulatory interactions or statistical artifacts.

6. The evaluation framework raises serious concerns about potential circularity. Reported high accuracy metrics derive from analyses using differentially accessible peaks, but performance on all accessible peaks remains unknown. The use of other cell-type-specific CREs as background may also bias against broadly active regulatory elements. More rigorous benchmarking is needed.

7. The model's feature representation is overly simplistic, ignoring well-established factors like binding affinity (as measured by ChIP-seq intensity or motif matching scores) and cooperative interactions between TFs. These omissions are particularly concerning given extensive literature demonstrating their importance for accurate cell-type classification.

8. The biological validation experiments are insufficient to support the conclusion. The synthetic enhancer tests only show that SHAP top motifs can drive activity, but can not demonstrate that motif combinations alone (without considering spacing or order) determine specificity. The cross-species predictions also lack proper functional validation. These experimental gaps make it unclear whether the results reflect true biological mechanisms or just artificial experimental conditions.

9. Benchmark comparisons are problematic as most tested algorithms (DNAbert/Enformer) are designed for long-range regulatory contexts. The fixed 500bp windows may artificially truncate important regulatory elements like super-enhancers, putting these methods at an unfair disadvantage.

10. Benchmark against at least three state-of-the-art motif or ChIP-seq-based methods.

11. The use of only differential peaks introduces prior knowledge into the classification task. The performance on all accessible peaks needs to be reported to properly assess the method's generalizability.

Minor Concerns:

12. Potential feature redundancy exists from similar TF family motifs in motif databases, which may affect model performance.

13. The choice of background set (other cell-type CREs) may introduce bias against broadly active regulatory elements.

14. For certain cell types with limited data such as plant cells with <300 CREs, there are concerns about model robustness and potential overfitting.

15. The justification for using $q < 0.5$ as the FIMO threshold is lacking and should be explained. Results of different thresholds should be presented.

16. There are obvious errors in figure legends such as Figure 4B that need correction.

(Remarks on code availability)

To enhance reproducibility, the authors should provide more detailed data processing steps in the GitHub repository.

Version 1:

Reviewer comments:

Reviewer #1

(Remarks to the Author)

I am happy with the updates and I recommend the manuscript for publication.

(Remarks on code availability)

Reviewer #2

(Remarks to the Author)

The authors have directly and appropriately addressed all my concerns from the first submission. Discussion of prior literature is greatly improved, placing the work and its novelty in better context. Limitations are acknowledged. And several other smaller improvements were made. I now recommend acceptance.

(Remarks on code availability)

Reviewer #3

(Remarks to the Author)

(Remarks on code availability)

Reviewer #4

(Remarks to the Author)

The authors have addressed most concerns with additional data. However, critical limitations of sequence/motif-based methods (the inability to distinguish TF family members) were dismissed as field-wide problems rather than substantively addressed.

(Remarks on code availability)

Authors have significantly improved the reproducibility of the code.

REVIEWER COMMENTS

Reviewer #1 (Remarks to the Author):

Here, Cornejo-Paramo et al present Bag-of-motifs (BOM), a novel computational framework for learning cell-type specific regulatory activity of distal enhancer sequences. Decoding regulatory elements and identifying TF binding motifs and their impact on gene expression remains one of the major challenges in computational biology. Today, there are large volumes of data available profiling open chromatin, TF binding, and gene expression at single cell resolution. Yet, there is still a need for improved computational methods as existing ones, including deep learning based ones, have so far not become widely used towards identification of cell type specific regulators.

In light of these challenges, I found the current manuscript a welcome contribution. The accuracy achieved by their classifier compared to state-of-the-art methods is highly impressive, and BOM has the potential to help researchers pinpoint key motifs when dissecting distal enhancers. Nevertheless, I believe that the following major issues must be addressed before the manuscript can be accepted for publication:

Overlapping TF binding motifs. Why do overlapping TF binding motifs need to be removed as outlined in the methods? It is assumed that the higher-affinity motifs are more likely to bind. Is this supported by any literature? The authors should justify their approach more. Given their note on relaxed motif detection thresholds (L779), this part of the algorithm looks like a step in the opposite direction. Also, how stable is the outlined approach? For example, if the sequence is scanned from the end instead of the beginning of the CRE, how different are the results?

Thank you for the feedback. To clarify, we did not remove overlapping motifs when creating our matrices for the models, as overlapping motifs offer important insights and enhance predictive accuracy, as the reviewer noted.

We assume the reviewer was referring to our methods, where we described the process of motif removal. We apologize for any confusion; this section is meant to evaluate the impact of motif removal, which was included in the supplemental results and might have been overlooked. This was done to formally assess the effect of including overlapping motifs in the count matrices. We have moved this to a more prominent position in the main results, in the section titled "Lenient motif detection thresholds improve performance."

In this section, we tested a range of q-value thresholds for motif calling using FIMO ($q \leq 0.1, 0.3, 0.5$). Stricter thresholds decreased the predictive performance of our models, indicating that inclusion of weaker, potentially degenerate, or lower-affinity motifs provides important information for cell-type-specific prediction (**Fig. S6, Table S12**).

We compared FIMO-predicted motifs to TF binding data for six key cardiac TFs using bioChIP-seq. Importantly, for TFs such as Nkx2-5 and Tead1, only a relaxed q-value threshold (>0.1) recovered binding sites overlapping ChIP-seq summits (**Fig. S7**). This demonstrates that a substantial fraction of bona fide binding events would be missed with overly stringent motif thresholds, validating our approach for motif detection.

Reducing motif counts by even a modest amount ($>10\%$) led to over a 50% drop in model sensitivity (**Fig. S8**). Eliminating overlapping motif annotations—a commonly proposed strategy to reduce false positives—also led to a substantial reduction in classifier performance ($\Delta\text{auROC} = 0.32$), suggesting that overlapping and weaker motifs may encode meaningful combinatorial or secondary motif interactions.

Regarding the stability of the results, FIMO will test a PWM in both forward and reverse directions, so the direction of the input sequence is not important. We have added this information to the manuscript.

CRE numbers. There is a substantial variation in CRE numbers (Table S1). Is this a technical artifact, or are there some biological reasons for the variation observed? Is having fewer CREs degrading the performance of the model?

The number of CREs that can be identified depends on how many cells of that type are available. Therefore, rarer cell types tend to have fewer CREs identified. Having fewer CREs can affect the model's performance for that cell type. In our code, we allow users to downsample CREs to balance the dataset so that the positive set (CREs of the cell type of interest) and the background (other CREs) are similar in size, ensuring that performance metrics accurately reflect the model's capabilities.

To give more guidance for users, we subsampled CREs from the endothelium CREs in the E8.25 dataset, repeated BOM training and predictions to show how the number of cells affects model performance. The Matthews Correlation Coefficient (MCC) stayed above 0.7 when $n > 330$ (**Fig. S2b, Table S7**). Even with just 30 CREs, we saw reasonable success in identifying endothelium-specific CREs (MCC=0.7). But, this can vary depending on the cell type and background used. We recommend using at least 100 CREs for running BOM. We have added this information to the manuscript.

Negative controls. I would like to see additional negative control experiments to ensure that BOM picks up biologically meaningful experiments. That is, how would the model perform if it were assigned the sequences 2 kb upstream of each CRE instead of the sequence overlapping the CRE?

Thank you for this suggestion. We have now done this (both up and downstream 2kb). We performed this analysis for 6 cell types. BOM showed a good performance in distinguishing the cell type-specific CREs from flanking sequences (false negative rate = 0.01 – 0.29). We have added this to our manuscript in **Fig. 1e, Fig. S2b, and Table S7**.

I understand that there are space constraints, but I was wondering if Fig 5c could be redesigned to use bars instead of a heatmap? As it stands I find it very difficult to compare the three models. Maybe my colorvision is not sharp enough, but for Fig 5d I can only distinguish two colors, corresponding to the two extremes of the scale. Is it the case that there are no real intermediate values? If so, could the authors consider displaying the information differently? Also, are these supposed to be violin plots? I do not understand what the blue boxes and red lines represent.

Thank you for the suggestion. We have changed the plot style to bar plots to make it easier to understand intuitively (Figure 5d) and emphasized the value of interest in 5c with a bold outline.

Testing the method with the provided example in the tutorial.md was not easy. We had two people testing it with different levels of experience. First, the fimo command line in the tutorial gave different output, and results ended up without genomic coordinates. Running it through R worked okay with the sample input. However the output was in ensembl whereas in the tutorial is Gencode. Therefore, we could not reproduce the same output. In general, mixing of ensembl and gencode annotations is confusing, although it does not make a difference in practice. Ideally, the method should clearly outline how to deal with bed files originating from Ensembl or gencode annotations. This will facilitate the wider adoption of the tool with non-computational people.

Thank you for the suggestion. The different output from the FIMO issue is due to a recent update of the FIMO software. We have made sure that the code is now compatible with the latest

version and have specified the version number in the GitHub repository. We have fixed the output so that it is consistent with Gencode convention. All bed files are now checked to make sure that it is reformatted, if needed, to the Gencode style of annotation ensuring consistency with the tutorial.

Also, the following minor issues need to be addressed:

L83: For the multiclass model, could the authors also provide a confusion matrix to visualize what misclassifications are the most common?

We have now provided this for the primary dataset analysed (mouse E8.25) and added it to the supplementary material (**Fig S3**).

L85: The authors mention a latent space embedding, but I could not find any information in the methods about how this was done. Could the authors please clarify?

The latent space embedding was performed using cisTopic. We have now clarified this in the Methods.

L170: "sequences could not be unaligned" What does it mean to unalign a sequence (is this even a real verb?)? I am not familiar with this procedure, so please clarify.

Thank you for pointing out this mistake. We have now fixed this. It should be that the sequences could not be aligned between the genomes using the UCSC liftOver tool.

L390: Filtering-based Motif Similarity. Not enough information is provided on how this filtering happened. How is each motif represented? As PWM? How were different length motifs compared?

This clustering was performed using Gimmemotifs (Heeringen and Veenstra 2011). We have added more details to the manuscript. Each motif is represented as a PWM. When comparing motifs of different lengths, the weighted information content similarity metric is calculated for all combinations of various lengths, and the highest similarity score is recorded. Then, an iterative clustering method is applied to group similar motifs and reduce redundancy.

L292: I could not source reference 50 of the mouse adult heart.

This work is currently unpublished, but we have now added the data to the Zenodo repository with the following DOI (DOI: 10.5281/zenodo.15720256) (<https://zenodo.org/records/15720256>).

L403: No investigation of overlapping binding motifs occurred. The authors should correct the sentence.

We have moved this results section to the main text from its previous supplementary location, as noted above.

Reviewer #1 (Remarks on code availability):

Please see above

Reviewer #2 (Remarks to the Author):

This article presents a machine learning method, Bag-of-Motifs (BOM), for learning combinations of motifs that identify cell type specific enhancers. The method is plausible (if fairly straightforward) and shows success on a large and diverse collection of problems to which they apply it. Moreover, they perform wetlab work to confirm some predictions of functional enhancers. The main downside of the paper is that they fail to recognize much prior literature and some prior methods relevant to their work - presenting their contributions as if they are much more novel than they really are, and potentially not comparing their method to other methods that might also perform well. Although there is some novelty and considerable effort represented in the work. In more detail, the strengths and weaknesses of the article are as follows.

We have expanded our introduction to better place our contributions in the context of previous work. Our supplementary table (previously **Table S21** now **Table S1**) explains BOM's contribution relative to popular methods. We discuss some popular methods, including IMAGE, ISMARA, LS-GKM, MEDEA, Gimme maelstrom, and ChromVAR. This list is by no means exhaustive, and we have generally selected methods based on their representative algorithm and the popularity of the tool. The table aims to clarify how these models compare in function to BOM, thereby better defining the gap that BOM can fill. We have also referenced methods in deep learning, which do not explicitly use motif vectors but attempt to model enhancer activity.

1) The basic algorithm they propose is that they take as input single-cell or single-nucleus ATAC-seq data, from which cell types are identified, and differentially accessible chromatin regions are identified for those cell types. They filter out regions that are too near genes. They count TF motifs in those regions, create a vector representation for each region. Then they use XGBoost to predict the cell type specific regions, and SHAP to evaluate the importance of particular motifs. This all makes sense, algorithmically and biologically.

2) However, the authors really ought to cite more prior work to show that these ideas are not coming from nowhere. For instance, representing promoters (and to a lesser extent enhancers) by vectors of motif counts has been done for decades - ever since the early post-human-genome days. There's dozens of such papers at least - too many for me to list. But the point is that representing the potential regulatory- or condition-specific character of genomic regions by vectors of motif hits has a long history. The authors should make more of an effort to find such works, cite them, and possibly even compare those methods to theirs. (Applying this idea to sc/sn-ATAC-seq regions was not done back then, of course, because such data only exists in recent years!) Indeed, the four main components of their algorithm -- differential accessibility, TF count vector representation, XGBoost, and SHAP -- are all individually established in genomics, including in the study of enhancers. The authors need to cite more of that prior work, even if/though the particular way that they have combined those ideas in the present paper has not been done before.

As described above, we have now included a comprehensive review of how motif counts are used to identify regulatory sequences and compare enhancers across different conditions. We highlight that our main focus is on evaluating the effectiveness of our pipeline in distinguishing regulatory sequences between conditions. Additionally, we have added a section on snATAC-seq to our introduction, recognizing that, along with new computational methods and improved PWM databases, these advances have enabled us to better define cell-type specific chromatin signatures across complex tissues and entire organisms. We also note that the high performance of BOM results from recent algorithmic developments, and we continue to reference these methods, including XGBoost, SHAP, and Gimmemotifs, throughout the manuscript.

3) Another strength of the paper is its successful demonstration on a large number of diverse examples. They look at cell types in mouse embryonic development, human cell lines, zebra fish, human blood, drosophila, arabidopsis, cross-species analysis of cardiomyocytes, etc. Personally I

would have liked to see Supp fig 4 about AML motifs in a main figure. But this is a small point. The authors have done considerable work to acquire these datasets, run their approach and some competing approaches, and intelligently analyze/interpret their results.

Thank you. We have added Supp Fig 4 about AML motifs to the main figure **Fig 3d**.

4) A considerable strength is also the effort to move beyond the drylab to do wetlab experiments. The authors demonstrate that they can add to an enhancer cell type specific TF motifs that they detected, and that these drive gene expression in a cell type specific manner. While synthesizing and testing enhancers in vivo is fairly straightforward for appropriately equipped wetlabs, few primarily-computational papers make any such effort (or even have the capacity or collaborators to do so). I would like to know the exact enhancer sequences they tested. I'm not sure if they're reported anywhere in the paper. (I looked but may have missed them.) They could even be potentially show in the main figure - instead of the boring grey bars, they could show exactly where the motifs were inserted in the enhancer each time. Nevertheless, this is a strength of the paper. A deeper analysis might look at the expression levels of the TFs whose motifs they inserted, and see which ones are expressed. And a yet deeper analysis might look at different combinations of TF motifs. But already the experiments represent a validation of their approach.

Thank you for pointing this out. The sequences and motifs inserted are listed in a supplemental Excel file; however, we forgot to reference it in the appropriate section. This has now been corrected. As suggested, we also revised our figure showing the relative location of each motif in the main figure (**Fig 6**).

5) As mentioned above, a general failure of the paper is to recognize, cite, or utilize a large body of previous work on enhancer motifs, or to even recognize the state of the art in that field. Moreover, there are too many unsubstantiated claims. For instance, consider the first line of the Discussion: "Our results implies that motif composition, rather than primary sequence conservation, plays a central role in regulatory evolution, providing new insights into how gene regulatory networks adapt and diversity." Firstly, it has already been known for decades that motif composition is conserved far more than primary sequence, and evidence of this continues to roll in as ever more genomes are sequenced and compared. It is an appalling disservice to previous work to state this as if the present paper is discovering this newly. Moreover, the present paper does not study the evolution of regulation in any substantial way. There is only one comparison of mouse versus human enhancer motifs in cardiomyocytes - far from a comprehensive evaluation of regulatory evolution. Further, the performance of their system when trained on one species and tested on another is substantially worse than when trained and tested on the same species. So it is not clear how strongly, in a quantitative performance sense, evolution is conserving motif composition in the sense that they study it. Finally, I would question precisely what 'new insights' this paper offers into regulatory network evolution. I would suggest a thorough reading of, among other things, work from the Zoonomia consortium and Andreas Pfennig and various colleagues in particular (including his TACIT algorithm), to better understand the state of the art in studying the evolution of regulatory elements, and to place the present work in more proper context.

Or, take the second line in the Discussion "Traditional methods for identifying regulatory conservation rely on sequence alignment, which often fail to detect functionally similar regulatory elements lacking strict sequence similarity." I am not sure precisely what methods they are referring to. Certainly it is normal to look for orthologous regions when studying a new species. However, it has again for decades been recognized that this is insufficient for regulatory regions. Indeed, it is often insufficient even for genes, where we know 3D structure of proteins is far more conserved than primary sequence. Thus, there is a whole collection of tools that one can use to study and try to establish links between regulatory regions in difference species. (Again see Zoonomia consortium, but many others as well!) If one is talking within species, then TF motif enrichment analyses have long been one of those tools and continues to be a key approach. For

instance, the MEDEA paper looks at TF motif enrichment in cell type specific differentially-accessible regions - very similar in spirit to the current paper, but not cited. And that is but one example.

We have acknowledged this criticism and thoroughly revised the Discussion. We agree that our results do not directly demonstrate a study of enhancer evolution and have removed that section. We highlight our findings in relation to BOM's high performance in predicting and classifying CREs across different conditions, benchmarking it against some of the most widely used methods, including recent deep learning approaches. The ability to identify such sequences accurately also enables further exploration of topics such as generative, cell-type-specific enhancers. Our work also provides a valuable comparison point for newer deep learning models, which we expand upon in the Discussion.

We also referenced and compared other methods, such as MEDEA, and included them in our Introduction. We cite TACIT in the section on sequence conservation, noting that TACIT focuses on chromatin accessibility conservation rather than sequence conservation, while BOM is used to explore this from a motif perspective. The models perform less well across species than within species. This is expected because different genomes have different nucleotide compositions, and we did not account for this bias in our comparison. The main point of the section is that a motif-based strategy, such as BOM, can perform better at identifying functional similarity (e.g., cell-type-specific chromatin accessibility) compared to traditional alignment strategies, like BLAT. However, a targeted approach for studying evolutionary conservation is outside the scope of this manuscript, as such a method would also benefit from incorporating conserved synteny, model training using data from multiple species, and genome bias corrections.

6) An important but easily remedied criticism of the work is that they refer to gene regulation far too much. They study differentially accessible regions (DARs). Such regions *may* have a regulatory function or they may not. Some estimates suggest that many do not, and even that many H3K27ac-associated regions (which is a classic enhancer mark) are not truly regulatory but mere side effects of promiscuous marking or 3D chromatin conformation bringing together non-regulatory regions near marked regulatory regions. With the obvious exception of the final wetlab portion of their study, very little of what they study is definitively regulatory, but merely accessible, and they should change the language accordingly.

We acknowledge that the gold standard for identifying a sequence as an enhancer is activity testing, preferably *in vivo*. However, this is difficult to perform in high-throughput. We have relied on conventions by labelling specific histone-marked regions and distal open chromatin regions as enhancers. We have added the terms 'candidate' or 'putative' enhancers to indicate sequences that have not been tested for enhancer activity.

Reviewer #2 (Remarks on code availability):

I checked that the repository is there and browsed the code to some extent, but did not download and test.

Reviewer #3 (Remarks to the Author):

Reviewer #4 (Remarks to the Author):

This manuscript introduces BOM (Bag-of-Motifs), a computational framework that utilizes transcription factor binding motif composition to predict cell-type-specific regulatory elements. The method employs an XGBoost model trained on motif occurrence counts and demonstrates applications across multiple species and biological contexts. While the feature attribution approach for identifying key transcription factors presents an interesting perspective, fundamental issues regarding the algorithm's innovation, biological purpose, experimental validation, and comparative analyses significantly undermine the study's conclusions. In its current form, the manuscript is not suitable for publication. The following concerns should be comprehensively addressed to support the research claims:

Major Concerns:

1. The core methodology closely resembles the established chromVAR framework in its use of motif composition for regulatory analysis. The authors must explicitly demonstrate how BOM provides meaningful improvements over chromVAR's motif accessibility deviation approach, which was not cited.

Our supplementary table (formerly Table S21, now **Table S1**) explains BOM's contribution relative to popular methods, including IMAGE, ISMARA, LS-GKM, and ChromVAR. We have now added a comprehensive introduction that compares these methods to the Introduction. Compared to BOM, ChromVAR does not provide performance metrics to understand how well the model generalizes to unseen instances. It also lacks the ability to interpret individual CREs.

2. The study claims to predict cell-type-specific regulatory elements (CREs), but such elements are already reliably identified through standard experimental methods (e.g., differential ATAC-seq peak calling). The authors must better justify the need for computational prediction when experimental assays can directly identify these regions.

We thank the reviewer for their comment, but we believe this concern reflects a misunderstanding of the purpose and premise of the motif-based classification models such as BOM. Experimental assays such as ATAC-seq, DNase-seq, or ChIP-seq are indeed effective for identifying accessible chromatin regions and for mapping cell-type-specific open chromatin landscapes. However, these assays do not provide mechanistic insight into *why* a given region is cell-type-specific.

Motif-based computational models, including BOM, address a fundamentally different question: Can we predict the cell-type specificity of CREs from their underlying sequence alone? This is not a substitute for differential peak calling. Rather, it is an effort to model the sequence context using motif combinations that determine cell-type-specific regulatory activity.

This approach has many precedents, as seen in the development of models such as gkm-SVM, DeepSEA, Enformer, and numerous motif-based classifiers, all of which address questions not answerable by differential peak calling alone.

Furthermore, unlike many motif-based tools that test for the statistical enrichment of motifs, BOM compares predictive ability. Predictive ability assesses whether sequence features are sufficient to generalize, which is key for both mechanistic understanding and practical applications, such as guided enhancer design (**Fig 6**).

We have provided a more detailed introduction to place BOM in the context of the work previously done in the field of research, to clarify our overall purpose.

3. The manuscript suffers from a profound conceptual disconnect between its stated objectives and actual implementation. While the title and Results sections repeatedly claim that BOM predicts "cell-type-specific regulatory elements" (e.g., Result 1: "A motif-based framework..."),

Result 2: "BOM outperforms..."), the methodology in fact operates in the opposite direction: it uses experimentally pre-defined cell-type-specific CREs (identified through conventional differential analysis) as input to predict cell type labels.

Supervised learning methods require labelled training data. Without this, it is unable to predict. Most machine learning models in regulatory genomics follow this paradigm.

After training, BOM operates on previously unseen DNA sequences. The process involves using experimental data to define classes, training models on sequences to learn features, and making predictions on new data. However, it is important to note that we do not feed all the data into training and keep a random subset for evaluation (validation and testing) to understand how the model generalizes. This subset of the data is held out, meaning that the testing and fine-tuning of performance are done on data that the model has not seen before.

We then use this learned model and employ an explainable AI method called SHAP to identify the most informative motifs for distinguishing between conditions.

4. The motif-based classification approach fundamentally fails to account for differential activity of transcription factor (TF) family members across cell types. While factors like BCL11A and BCL11B share similar binding motifs, they exhibit distinct regulatory roles in different immune cell differentiation lineage. The current binary motif representation cannot capture these functional differences, potentially leading to misclassification of cell states/phenotype where TF activity is modulated through post-translational modifications rather than solely binding site number.

We thank the reviewer for this thoughtful point, which highlights a general limitation of all sequence-based motif classification models. We agree that many TF families recognize highly similar or even identical binding motifs. As a result, a model based solely on sequence cannot, in principle, distinguish which specific TF family member is active, nor can it account for regulatory specificity arising from differential TF expression, cofactor availability, or post-translational modifications.

Our approach, like all motif-based or sequence-based classifiers (including gkm-SVM, DeepBind, DeepSEA, Enformer, etc.), models the potential encoded in the DNA sequence for cell-type-specific activity. It is important to note, however, that it employs multiple motifs to achieve this, not just one, and that our model allows these motifs to be combined in a non-linear and flexible manner. This is important as similar members of the motif family are likely to interact with different cooperatively binding TFs.

We also demonstrate that one can bring in additional sources of information, such as gene expression data where available, to supplement the information provided by DNA (**Fig 5e&f**)

5. The motif scanning using FIMO may introduce substantial false positives in TF binding predictions. Authors should validate their predictions by leveraging large-scale TF ChIP-seq datasets to demonstrate that identified motifs correspond to binding events in relevant cell types. Without such experimental validation, it remains unclear whether predicted motif combinations reflect genuine regulatory interactions or statistical artifacts.

We appreciate the reviewer's concern regarding potential false positives in motif scanning and the importance of validating sequence-based predictions with experimental data. To address this, we systematically explored motif detection thresholds and benchmarked our predictions against large-scale bioChIP-seq datasets.

As mentioned above, we tested a range of q-value thresholds for motif calling using FIMO ($q \leq 0.1, 0.3, 0.5$). Stricter thresholds decreased the predictive performance of our models, indicating

that inclusion of weaker, potentially degenerate, or lower-affinity motifs provides important information for cell-type-specific prediction (**Fig. S6, Table S12**).

We compared FIMO-predicted motifs to TF binding data for six key cardiac TFs using bioChIP-seq. Importantly, for TFs such as Nkx2-5 and Tead1, only a relaxed q-value threshold (>0.1) recovered binding sites overlapping ChIP-seq summits (**Fig. S7**). This demonstrates that a substantial fraction of bona fide binding events would be missed with overly stringent motif thresholds, validating our approach for motif detection.

Reducing motif counts by even a modest amount ($>10\%$) led to over a 50% drop in model sensitivity (**Fig. S8**). Eliminating overlapping motif annotations—a commonly proposed strategy to reduce false positives—also led to a substantial reduction in classifier performance ($\Delta\text{auROC} = 0.32$), suggesting that overlapping and weaker motifs may encode meaningful combinatorial or secondary motif interactions

In summary, we validated our motif discovery and selection approach using independent ChIP-seq datasets and found that including a broader set of motif matches is critical both for predictive performance and for recovering bona fide TF binding sites. While false positives are an inherent limitation of motif scanning, our results demonstrate that our model is robust, empirically validated, and reflects a generalized view of the biological complexity of regulatory element architecture. These results are in “Lenient motif detection thresholds improve performance.”

6. The evaluation framework raises serious concerns about potential circularity. Reported high accuracy metrics derive from analyses using differentially accessible peaks, but performance on all accessible peaks remains unknown. The use of other cell-type-specific CREs as background may also bias against broadly active regulatory elements. More rigorous benchmarking is needed.

We respectfully disagree with the assertion that our evaluation framework is subject to circularity. Our goal is to understand the sequence basis of cell-type specificity, which requires training and testing on differentially accessible peaks. During evaluation, we ensure strict separation of training and test sets. At no point are the features used for model input (motif annotations) used in defining the labels or in differential peak calling.

However, we included additional analyses using three benchmarking strategies to rigorously demonstrate model performance, minimize the risk of methodological bias, and demonstrate that our findings are robust across independent datasets. First, we trained and evaluated BOM in a multilabel setting on all open chromatin regions, which resulted in lower predictive performance as expected (mean auPR = 0.44, mean F1 = 0.3, mean MCC = 0.32; **Fig. 1f**), indicating the challenge of predicting broadly active CREs from sequence alone which we discuss in the text. Second, we benchmarked BOM on non-accessible regions using flanking regions to assess and confirm general specificity and low FPR (**Fig. 1e, S2b, Table S7**). Third, we assessed model generalizability using an independent snATAC-seq dataset from another laboratory where the embryonic time point was also slightly different (E8.5 rather than E8.25), where BOM maintained high predictive accuracy in an independent dataset despite a difference between the embryonic timepoints (**Fig. 1g, Table S10**).

7. The model's feature representation is overly simplistic, ignoring well-established factors like binding affinity (as measured by ChIP-seq intensity or motif matching scores) and cooperative interactions between TFs. These omissions are particularly concerning given extensive literature demonstrating their importance for accurate cell-type classification.

Weaker binding affinities of individual TFs can be captured by adjusting the detection threshold, as discussed above. As detailed above, we have systematically tested a range of motif match score thresholds and selected those that optimized predictive performance and concordance with ChIP-seq-validated TF binding sites. This calibration process allows our model to incorporate a spectrum of motif affinities, including lower-scoring and potentially degenerate sites that may contribute to regulatory specificity. However, weaker regulatory elements, with weaker TF binding sites that are less well-characterized, may be generally less accurately detected by experimental methods, but also less accurately predicted based on motif scoring and subsequent BOM modeling. However, this is not a problem unique to BOM.

Cooperativity between TFs should also generally be captured, though not explicitly explained, similar to binding affinity. This does not mean that motif affinity and ordering are unimportant. However, understanding those subtle interactions, which can be highly locus-specific, requires further analysis and is outside the scope of the tool. BOM is designed to explain the most significant differences between two groups, cell types, or conditions, helping researchers identify the most informative set of transcription factors (TFs).

8. The biological validation experiments are insufficient to support the conclusion. The synthetic enhancer tests only show that SHAP top motifs can drive activity, but can not demonstrate that motif combinations alone (without considering spacing or order) determine specificity. The cross-species predictions also lack proper functional validation. These experimental gaps make it unclear whether the results reflect true biological mechanisms or just artificial experimental conditions.

Our results provide functional support for the relevance of the identified sequence motifs. The synthetic enhancer experiments are designed primarily to test the functional sufficiency of SHAP-identified top motifs for driving enhancer activity in a given cell type, not to address the roles of motif spacing or order directly. Our design—randomly dispersing motifs—assesses whether motif presence alone is predictive of cell-type activity. Additional work is needed to systematically test motif arrangement and combinatorial effects, but this is outside the scope of this work, and we do not claim that motif arrangements are not important, only that they are, not required for delineating well-defined cell types. Other projects have focused on systematically assessing combinatorial motif effects using MPRA assays (e.g. Agarwal et al. 2025 Nature).

Similarly, our current results are used to demonstrate the predictive power of cross-species predictions. We disagree that we lack proper functional validation. The results used to calculate prediction metrics are experimentally validated using *in vivo* data. Certainly, to properly assess enhancer function, *in vivo* enhancer activity testing would be required; however, this is outside the scope of this work, and we are careful not to claim that these regions function as enhancers.

9. Benchmark comparisons are problematic as most tested algorithms (DNAbert/Enformer) are designed for long-range regulatory contexts. The fixed 500bp windows may artificially truncate important regulatory elements like super-enhancers, putting these methods at an unfair disadvantage.

We appreciate the reviewer's point regarding potential benchmarking biases arising from window size. We agree that truncating the sequences could impact model prediction, as the status of open chromatin, histone marks, and enhancer activity can be influenced by regions beyond the sequence used by BOM. We have added this as a limitation of our model in the Discussion.

However, to specifically test whether BOM's performance was confounded by super-enhancer context (where window truncation might be most problematic), we stratified CREs by whether they overlapped super-enhancers or not. We found that BOM's accuracy was similar for both

super-enhancer and non-super-enhancer CREs, suggesting that BOM's window size is not driving its high performance (**Fig. S5, Table S11**).

We also would like to clarify that DNABERT is not designed to capture long-range regulatory interactions; rather, it is specifically trained on short fixed-length DNA sequence windows and there is no regulatory data used to train the model. In addition, recent evidence (e.g., work by Julien Gagneur and colleagues) demonstrates that, for models like Enformer, performance on long-range regulatory prediction remains limited, and that prediction accuracy is often highest for more proximal regulatory elements.

We agree that future benchmarking efforts, ideally with models run on their optimal input window sizes, would further clarify the relative advantages of long-range models like Enformer in the context of super-enhancers and other extended regulatory regions.

10. Benchmark against at least three state-of-the-art motif or ChIP-seq-based methods.

We believe that current comparisons provide the most broadly relevant performance benchmark for BOM. We believe investigating different methods such as, DNABERT and Enformer, two state-of-the-art deep learning frameworks for regulatory sequence prediction, is of greater general interest than comparing with more similar methods, given BOM's already high predictive ability. DNABERT and Enformer along with gkm-SVM, represent the most interesting and competitive available approaches for sequence-based prediction and have become standards for evaluating regulatory sequence models.

We note that Enformer is trained on ChIP-seq data, in addition to other regulatory datasets. DNABERT and gkm-SVM can also be classed as state-of-the-art motif methods. We recognize that other models explicitly use TF motifs, and we have carefully surveyed the current landscape of motif-based and ChIP-seq-based prediction methods (**Table S1**). But we have focused specifically on those that represent the current frontier of machine learning-based regulatory sequence models.

We also note that many motif-based tools in the literature focus on enrichment analysis, and do not implement sequence-based predictive modeling or provide standardized metrics for direct performance comparison with our approach (**Table S1**).

11. The use of only differential peaks introduces prior knowledge into the classification task. The performance on all accessible peaks needs to be reported to properly assess the method's generalizability.

To clarify, our method does not aim to predict whether regions are accessible or inaccessible. That is a simpler task that has been thoroughly studied before (e.g., Lee et al 2011 Genome Res). Our goal is to identify the elements that define cell type identity, which requires training on differentially accessible peaks. As mentioned above, we have trained a multilabel predictor for all accessible regions (**Fig. 1f**), and the prediction is more challenging due to the combinatorial possibilities (e.g., that enhancers active in cell types A, B, and C is being correctly predicted as such).

Minor Concerns:

12. Potential feature redundancy exists from similar TF family motifs in motif databases, which may affect model performance.

To address this issue, we use GimmeMotifs, which is a motif database that clusters motifs to reduce redundancies. We have provided a better description of this process in the Methods.

13. The choice of background set (other cell-type CREs) may introduce bias against broadly active regulatory elements.

Please refer to the response for point 7.

14. For certain cell types with limited data such as plant cells with <300 CREs, there are concerns about model robustness and potential overfitting.

We test our model on held-out datasets to prevent overfitting. The model's performance slightly declines with less data, but it remains relatively high. To help users, we have included additional information and a new analysis (**Fig. S2b, Table S7**). Even with just 30 CREs, we observed reasonable performance in identifying endothelium-specific CREs (MCC=0.7). We note that performance will vary depending on cell type and background. For future models, we limited our selection to those with at least 100 CREs.

15. The justification for using $q < 0.5$ as the FIMO threshold is lacking and should be explained. Results of different thresholds should be presented.

We have added a new section to the main text on these thresholds. Although the section was present in the previous version, it was likely overlooked, as it was in an awkward location within the Supplementary Results.

16. There are obvious errors in figure legends such as Figure 4B that need correction.

Thank you, we have fixed this.

Reviewer #4 (Remarks on code availability):

To enhance reproducibility, the authors should provide more detailed data processing steps in the GitHub repository.

We have provided a GitHub repository for code and metadata that is used to perform the analyses in this manuscript, https://github.com/ewonglab/BOM_manuscript_data_scripts, in addition to the code for BOM itself with an associated tutorial (https://github.com/ewonglab/BOM_package).